# Rapid heat discharge during deep-sea eruptions generates megaplumes and disperses tephra

Samuel S. Pegler[1✉] & David J. Ferguson [2✉]

Deep-marine volcanism drives Earth's most energetic transfers of heat and mass between the crust and the oceans. While magmatic activity on the seafloor has been correlated with the occurrence of colossal enigmatic plumes of hydrothermal fluid known as megaplumes, little is known of the primary source and intensity of the energy release associated with seafloor volcanism. As a result, the specific origin of megaplumes remains ambiguous. By developing a mathematical model for the dispersal of submarine tephras, we show that the transport of pyroclasts requires an energy discharge that is sufficiently powerful (~1-2 TW) to form a hydrothermal plume with characteristics matching those of observed megaplumes in a matter of hours. Our results thereby directly link megaplume creation, active magma extrusion, and tephra dispersal. The energy flux at the plume source required to drive the dispersal is difficult to attain by purely volcanogenic means, and likely requires an additional input of heat, potentially from rapid evacuations of hot hydrothermal fluids triggered by dyke intrusion. In view of the ubiquity of submarine tephra deposits, our results demonstrate that intervals of rapid hydrothermal discharge are likely commonplace during deep-ocean volcanism.

[1] School of Mathematics, University of Leeds, Leeds, UK. [2] School of Earth and Environment, University of Leeds, Leeds, UK. ✉email: s.pegler@leeds.ac.uk; d.j.ferguson@leeds.ac.uk

The vast majority of Earth's volcanism occurs underwater in the deep oceans at mid-ocean ridges (MORs) and seamounts (>500 m deep). Submarine magmatism accounts for >80% of the global volcanic heat flux[1] and facilitates important chemical-physical interactions between the crust and the oceans via seafloor hydrothermal activity. A significant but poorly understood aspect of this hydrothermal activity is the generation of massive (10–150 km³) ephemeral emissions of hydrothermal fluid known as "megaplumes" (or large-volume "event plumes")[2–7] (Fig. 1). Megaplumes are characterized by high ratios of heat to hydrothermal chemical components compared to the plumes produced by chronic hydrothermal vents such as black smokers[7]. Their total energy contents are within the range ~$10^{16}$ – $10^{17}$ J, comparable to the annual thermal output from a typical mid-ocean ridge hydrothermal vent field[2], implying extremely high rates of energy discharge.

The detection of megaplumes along MORs by physicochemical measurements in the water column has occurred both fortuitously during pre-planned surveys[2,3,6,8] and during rapid response cruises undertaken following the detection of geophysical evidence for submarine eruptions[9,10], such as seismic or hydrophone activity (see[5] for a review). Subsequent ocean floor surveys, when conducted, have provided evidence for contemporaneous eruptive activity[4,11] and megaplume creation appears to be linked in space and time with deep sea volcanic events[5]. Observed concentrations of labile chemical species in megaplume fluids, such as $H_2$[12] and dissolved Fe[13], generally indicate that the period of hydrothermal discharge was relatively brief and that megaplume formation is likely an ephemeral process, probably associated with transient magmatic events.

Despite the apparent link with active volcanism, the dominant process that forms a megaplume remains unknown. Several theories exist for the source(s) of megaplume heat contents and fluids. These include: a purely volcanic origin via heat transfer from erupted lava and volatiles[14,15]; a magmatic origin due to heating of pore fluids by intruded magma in a dyke[16]; or a hydrothermal origin via the rapid evacuation of existing intracrustal fluid reservoirs[2,7]. Differentiating between these is challenging because few observations of active deep marine eruptions exist. In particular, while models of the dynamics of megaplumes have suggested they form rapidly[17], little is known of the rates of energy or volume discharge feeding the plume during a seafloor eruption, of the primary source of the hydrothermal input, nor of the role of eruption dynamics on plume formation.

Another open question regarding seafloor volcanism is the process responsible for the dispersal of pyroclastic deposits in the deep ocean. Previously thought to be largely absent in deep marine settings, imaging and sampling of the seafloor at ridges[15,18–20] and seamounts[21–28] has revealed the presence of tephra over many km², typically comprising sub-cm shards of volcanic glass. Dispersal distances for these tephras are inferred to reach several km. Older tephra-bearing sediments recovered from sediment cores taken on the flanks of MORs also indicate similar dispersal scales[29,30]. These tephras provide evidence for explosive pyroclastic eruption styles[15,19], something that has traditionally been considered extremely rare due to the high hydrostatic pressure[31,32] (we note that some debate exists as to the potential for tephra to be generated by magmatic fragmentation of fluid magma[15] versus other brecciation processes such as thermal granulation[33] and/or hydrovolcanic fragmentation[28]). Indeed, pyroclastic eruptions have been directly witnessed at water depths exceeding 500 m at submerged arc[34,35] and rear-arc[36] volcanoes. Despite their widespread occurrence, the dominant process responsible for dispersing submarine tephra remains unknown.

With the exception of the pyroclastic deposit studied here (sampled and mapped by[15]), no detailed information exists on the distribution of submarine tephras around their source eruptive vent or fissure, the location of which is typically uncertain. As such, the development of an explanation for the primary mechanism of tephra dispersal, as well as the assessment of the possibility to invert submarine depositional patterns for paleo-eruptive properties (as is routinely attempted for subaerial tephras[37]), have remained unexplored.

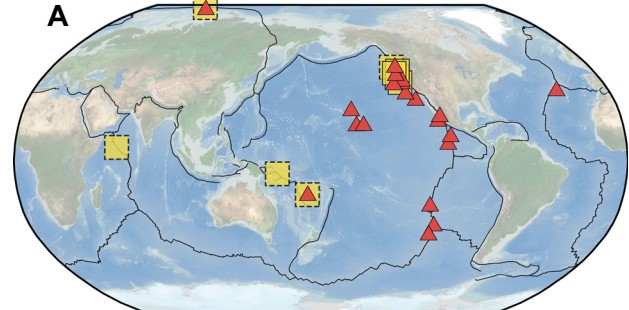

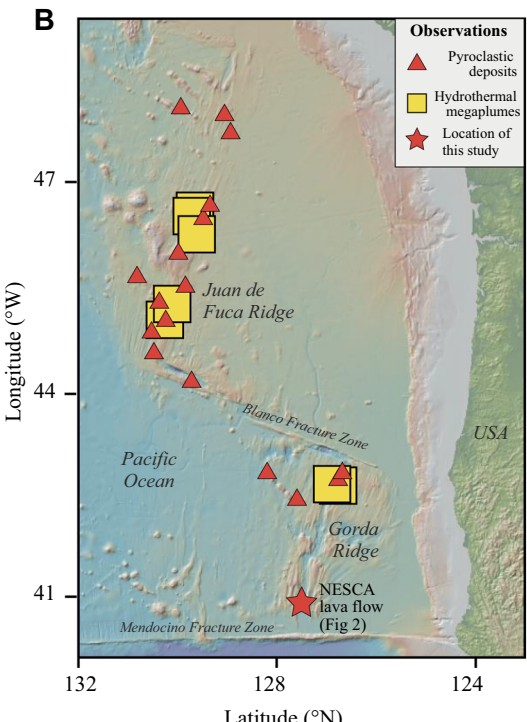

**Fig. 1 Observations of hydrothermal megaplumes and deep-marine tephra deposits.** Locations of megaplumes detected by water-column measurements (yellow boxes) and observed deep-marine pyroclastic tephras (red circles). Observations are shown (**A**) globally and (**B**) in the NE Pacific. Boxes with solid lines show plumes that have been mapped in three dimensions and therefore have known volumes (~10–150 km³; 7 observations), while dashed lines indicate those with chemical and physical characteristics consistent with a megaplume but without a confident volume estimate (5 observations). Deep-marine tephras have been discovered in multiple locations at both mid-ocean ridges (MORs) and seamounts. These encompass the global range of MOR spreading rates and water depths of up to 4 km. The preponderance of observations in the NE Pacific (shown in (**B**)) is related to the concentration of marine research in this region. The location of the eruption and tephra deposit used for our inversion (Fig. 2) is shown by the star symbol in (**B**). Tephra observations, particularly in (**B**), are from[15] with additional data from[18–24]. Megaplume observations are from compilations by[5,6]. Black lines in (**A**) show tectonic plate boundaries. The base map in (**B**) from geomapp.app.org[71].

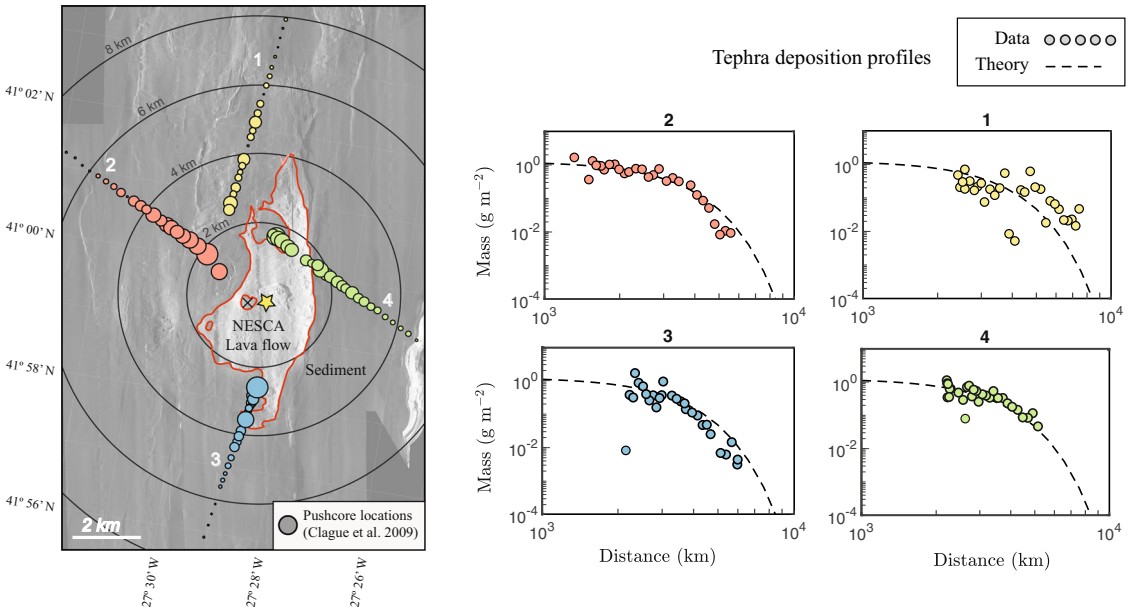

**Fig. 2 Map and tephra deposition pattern at the Northern Escanaba (NESCA) eruption site with theoretical fits.** Sidescan sonar imagery of the Northern Escanaba (NESCA) lava flow at the southern end of the Gorda Ridge (see Fig. 1A). Circles show locations of sediment pushcores taken along four profiles around the flow by[15], with the area of the data points proportional to the mass per unit area of tephra in the 250–500 µm range. The extent of the lava flow is indicated by a red outline. The plots on the right show the mass per unit area of the corresponding sampled data profiles (labeled 1–4) as a function of distance from an inferred center marked by the yellow star in the map. The inferred center is the unique position creating a global minimum in mean square error resulting from fitting our Gaussian model given by Eq. (1) to the data. This is ~800 m from the location of the actual eruptive vent, recently identified using high-resolution bathymetric data (D. Clague, pers. comm.) and shown by the black cross. The Gaussian dispersal pattern resulting from our regression analysis is shown as a dashed curve on each of the plots, for which the dispersal scale $L \approx 4.9$ km. The 15% error in estimating the mass of glass particles[15] corresponds to the approximate size of the circular markers. Concentric circles on the map represent 2-km increments from the calculated center. Sonar imagery from[38].

In this study, we formulate a model of particle dispersal by a hydrothermal plume, demonstrate that its predictions are consistent with observations of submarine deposits, and apply it to invert directly for the energy discharge rates produced during a submarine eruption. By further showing that the predicted energy contents align with independent oceanographic measurements of megaplumes, we establish a conclusive link between tephra producing eruptions and megaplume creation. Our model of buoyancy-driven tephra transport by the umbrella of volcanic plumes allows us to invert for the co-eruptive rate of energy discharge associated with a submarine hydrothermal energy release. The inference of the rate of energy release (independent of the total energy and volume content of observations) provides a new inroad to scrutinize proposed mechanisms of megaplume creation.

## Results

**Tephra deposit and dispersal characteristics.** The application of our model is possible owing to the existence of a unique dataset of tephra deposition from a single isolated submarine eruption, the basaltic Northern Escanaba (NESCA) lava flow[38], collected by D. Clague and co-workers[15] in the Northern Escanaba Trough at the Gorda Ridge, NE Pacific (Fig. 2). The lava and tephra from this eruption (estimated to have been emplaced around 300 years ago) are the only volcanic deposits in this region that overlie the sediments deposited by the Missoula floods[39]. Pyroclasts up to 1 mm in size were sampled and mapped around the lava flow via pushcores collected by a remotely operated vehicle, revealing lateral transport distances exceeding 5 km in all directions. The tephra particles were separated into size fractions and the relative mass of each fraction measured. The plots in Fig. 2 show the mass distribution of pyroclasts in the 250–500 µm range along each of

four profiles of pushcore data from[15] (the central position used to define the zero distance, shown as a yellow star in Fig. 2, will be discussed in our analysis below). In all cases, the mass of material decreases with distance from the source with an approximately axisymmetric, qualitatively Gaussian decay trend.

It has been suggested that the km-scale lateral transport of submarine tephras may be a consequence of vertical lofting above the eruptive vent, followed directly by settling within a sustained cross current[40]. Depending on location, the background flow in the deep sea forms from a superposition of deep ocean currents, tidal currents, mesoscale eddies, internal waves and turbulent mixing (e.g., breaking internal waves). In view of the near-axisymmetric form of the observed dispersal at NESCA, we propose that advection in a sustained cross flow is highly unlikely. Otherwise, the dispersal would be preferentially skewed in one direction. The lack of a dominant effect of sustained cross flows in the NESCA ash deposit is consistent with the absence of significant focused oceanic currents in the North East Pacific (month-long speed averages in this region of the deep ocean are <0.006 m s$^{-1}$ [41]). Sustained currents may affect submarine tephra dispersal within the localized areas of the ocean containing deep-ocean currents forming part of the global ocean circulation. Tidal currents could, in principle, produce a radial-like dispersal owing to their periodicity; however, by considering the trajectories of particles in a typical tidal field (see Supplementary Note 1), we determine that tidal reversal constrains the transport of tephra by tidal currents to a maximum distance of ~700 m from the eruptive source. In summary, the dispersal distances of >5 km observed in all directions at NESCA cannot be explained by advection within either deep-ocean currents or tidal currents.

We propose instead that the characteristics of the observed tephra deposition suggest a dominant transport mechanism by

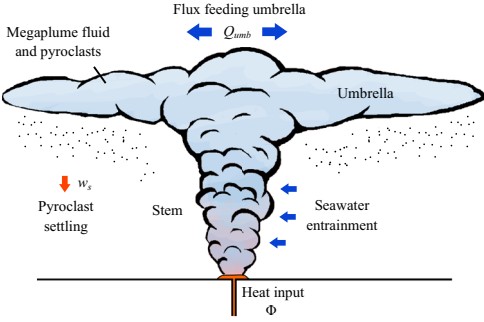

**Fig. 3 Schematic showing the configuration and the processes controlling buoyancy-driven submarine tephra dispersal.** The hydrothermal plume forms a turbulent convecting stem fed by lava heating and/or release of intracrustal fluid, which accumulates and cools following entrainment of ambient seawater. The stem feeds the neutrally buoyant umbrella, which forms a primarily horizontally flowing neutrally buoyant gravity current within the density stratification of the ocean, with a volumetric flux of $Q_{umb}$.

horizontal buoyancy-driven advection within the umbrella of a syn-eruptive hydrothermal plume. For an effectively quiescent ambient ocean, the umbrella of the plume will form an axisymmetric gravity current along a neutral level, advecting particles laterally by its own buoyancy-driven flow while maintaining the particles in suspension within turbulent eddies. Being denser than the water, the tephra particles will progressively fall from the suspension of the turbulent flow to produce a thinning deposition of particles in all directions.

Tephra transport in a buoyancy-driven plume umbrella is often considered in idealized prototypical fluid-mechanical analysis of tephra dispersal by subaerial eruptions[42,43]. However, it is neglected in standard models for inverting subaerial tephra data, owing to the need for a new kind of mathematical model to account for horizontal buoyancy-driven flow. The most standard models and inversion toolkits designed for subaerial eruptions[44,45] account for horizontal transport of particles via advection by atmospheric crosswinds and diffusive atmospheric mixing alone, thus neglecting the effect of advection by buoyancy within the plume umbrella. In situations where this approach is applied to near-axisymmetric subaerial eruptions, the method infers unphysical values for the atmospheric diffusivity[46], with the effect of buoyancy-driven flow in the plume umbrella attributable as a likely cause[47]. As we show here, buoyancy-driven spreading can generate substantial km-scale radial dispersal in the submarine context.

If the dispersal is, as we propose, advected by horizontal buoyancy-driven flow of the plume umbrella, then it should be possible to correlate tephra deposition distances directly with energy input rates. This correlation yields an inroad for the estimation of spatial, temporal and energetic characteristics of the heat discharge produced during volcanic eruptions. The analysis we present here demonstrates that a model based on horizontal buoyancy-driven transport predicts the characteristics of a natural tephra deposit, and we develop the first inversion of data based on these transport dynamics.

**Dispersal model and inversion method.** An input of heat at the seafloor will coalesce into a turbulent, primarily vertically convecting column of heated water, herein referred to as the stem of the plume (Fig. 3). This structure will both grow laterally and cool as it entrains ambient seawater[3]. Following an inertial overshoot, the plume will settle along a neutral level as a turbulent, primarily horizontally flowing neutrally buoyant gravity current (or intrusion), forming the umbrella of the plume. Tephra produced

during eruption of the lava will be carried by the plume stem into the umbrella, with some proportion of the tephra falling from the sides of the stem (a model predicting the tephra concentration through the stem of the plume and the proportion reaching the umbrella is developed in the Methods). On reaching the umbrella, the tephra will subsequently be transported primarily horizontally by buoyancy-driven flow within the umbrella, resulting in km-scale transport.

To develop our inversion methodology, we apply a model of the plume in two components, illustrated in Fig. 3. The stem of the plume is modeled as a turbulent, vertically convecting column of hot water, while the umbrella is modeled as a turbulent gravity current flowing along a neutral level of the ambient density stratification (see the Methods for details on these models). The two regions are coupled by a condition of continuous volumetric flux between the top of the stem and the radial origin of the umbrella at the neutral buoyancy level. Particles entrained into the plume will, following possible fallout from the stem (see Methods), propagate into the plume umbrella and settle from its base at a rate proportional to particle concentration[42,43]. The theory of particle settling from axisymmetric gravity currents[42] predicts a Gaussian deposit profile:

$$\Omega(r) = \Omega_0 \exp\left[-\pi(r/L)^2\right], \qquad (1)$$

where $\Omega(r)$ is the deposited particle mass per unit area of a group of particles (a size range) characterized by a particle settling speed of $w_s$, $L = (Q_{umb}/w_s)^{1/2}$ is a horizontal lengthscale representing the scale on which the mass of the tephra group decays (encapsulating ~93% of the mass of umbrella-deposited tephra of the particle group being considered), $Q_{umb}$ is the volume flux of fluid feeding the umbrella, and $\Omega_0$ is a constant representing the scale of accumulation (the total mass of tephra of the particle group being considered is $\Omega_0 L^2$). A derivation of the result of Eq. (1) is reviewed in the Methods. The dispersal lengthscale $L$ is independent of both the duration of the eruption and the rate of input of particles (either of which will only accumulate $\Omega_0$) and hence $L$ provides an independent fitting parameter that can be used to constrain the volumetric flux sourcing the umbrella via the formula:

$$Q_{umb} = w_s L^2. \qquad (2)$$

Since this inversion formula does not depend on the amount of tephra deposited (which is encapsulated in $\Omega_0$), even a small particle input can be sufficient to apply it.

The model above is based on a number of assumptions. First, Eq. (1) provides the deposition field under the assumption of a single (or representative) particle settling speed $w_s$. In our analysis, we will choose a specific particle range (extracted by sieving[15]) and assume a particle settling speed representing this group. Another assumption underlying the model is that the plume is sustained by an approximately steady buoyancy source over the eruptive duration. While a waning input rate can be anticipated under the various theories for megaplume creation, we can anticipate that an approximation of a constant input rate provides a representation of the averaged properties of the plume system during the main period of energy release. Attenuation of the source energy will provide a relatively smaller contribution to deposition at late times. A further assumption underlying the model is that the presence of particles does not significantly impact the fluid flow. By developing a model for particle transport in the stem of the plume (see Methods), we show that this is likely to be an excellent approximation for this application owing to the considerably larger proportion of plume fluid compared to particle mass.

Having determined the flux of fluid fed into the umbrella $Q_{umb}$ using Eq. (1), we utilize further mathematical models of the plume stem to relate this flux to the flux of heat energy sourcing the plume from the eruptive vent or fissure (whichever is the most appropriate source geometry for the eruption of interest) according to:

$$\Phi \approx \begin{cases} 0.187 \ k\left(\frac{N^5 Q_{umb}^4}{\varepsilon^2}\right)^{1/3} & (l \lesssim l_*), \\ 0.326 \ k\left(\frac{N^3 Q_{umb}^3}{\varepsilon l}\right)^{1/2} & (l \gtrsim l_*), \end{cases} \quad (3)$$

where $l$ is the source fissure length, $l_* = 3(\varepsilon Q_{umb}/N)^{1/3}$ is a lengthscale characterizing the fissure length on which the dynamics transition from axisymmetric to planar models, $\varepsilon \approx 0.1$ is the entrainment coefficient, $k \approx 2.1 \ \text{GW m}^{-4} \text{s}^3$ is the conversion factor between buoyancy flux and heat flux, and $N$ is the Brunt-Väisälä frequency of the ocean stratification. With the flux reaching the umbrella $Q_{umb}$ determined from the first stage of our inversion represented by Eq. (2), the formula above predicts the rate of hydrothermal heat input $\Phi$ introduced at the base of the plume necessary to generate this flux.

**Dispersal of the Northern Escanaba tephra.** The observed profiles of tephra deposition for the NESCA flow are shown for four groups of pushcores in Fig. 2, each forming an approximately linear path along the seafloor (Fig. 2). The deposition profiles all follow qualitatively Gaussian decay trends. This provides support for our essential hypothesis that buoyancy-driven flow in the umbrella was the primary driver of the dispersal, as represented by the prediction of Eq. (1). To estimate the dispersal scale $L$ and the center of the dispersal, $\boldsymbol{x}_c = (x_c, y_c)$, we determined the position $\boldsymbol{x}_c$ and values of $\Omega_0$ and $L$ (a total of four fitting parameters) that minimize the root mean square error:

$$E = \left(\frac{1}{N}\sum_{i=1}^{N}\left|\Omega_i - \Omega_0 e^{-\pi|\boldsymbol{x}_i - \boldsymbol{x}_c|^2/L^2}\right|^2\right)^{1/2}, \quad (4)$$

where $\Omega_i$ and $\boldsymbol{x}_i$ denote the mass per unit area and positions of particles in the sampled range of 250–500 μm of the $N$ pushcores. This was achieved by conducting a grid search of positions $\boldsymbol{x}_c$ over a rectangular region surrounding the lava flow. For each position on the grid, $\boldsymbol{x}_c$, the root mean square $E$ was minimized over values of $\Omega_0$ and $L$ using a nonlinear programming solver. The unique position yielding the overall minimum error, $\boldsymbol{x}_c = (-127.4892, 40.9893)$, is indicated by the yellow star in Fig. 2. The corresponding value of $L$ is 4.9 km. Recently acquired high-resolution bathymetry of the NESCA flow (D. Clague, pers. comm.) has revealed that the eruption occurred from a ring-fault around a small sediment hill indicated by a cross in Fig. 2, located ~800 m from our inferred center.

The fitted model of Eq. (1) determined from our global minimization is plotted alongside the data for each of the four pushcore groups in Fig. 2. There is consistent general agreement between the data and the Gaussian trends, with 51% of the data lying above the model curve, and an $R^2$ value of ~0.6. Bootstrapping $10^4$ resampled datasets yields a standard deviation of 400 m in the fitted value of $L$. Ours is a continuum model for the statistically averaged deposition field, and hence deviations between our model and the data are expected. We speculate that the scatter may represent syn- or post-depositional processes such as statistical noise in the turbulent and particle dynamics, the effect of topography (particles falling unevenly on sloped surfaces), sediment displacement and/or bioturbation, combined with a 15% predicted error in the measurement of the proportion of tephra in each pushcore[15] (indicated by the size of the markers in Fig. 2). Clague et al.[15] observed that the majority of the tephra

in their pushcore samples resided within the uppermost cm of the seafloor sediment, indicating that most of the sampled tephra had remained largely undisturbed since deposition. However, some particles had been mixed downwards by bioturbation, in some cases by up to 1 m, creating at least one source for outliers.

In addition to $L$, the other parameter required to complete our inversion for the umbrella flux using Eq. (2) is a representative settling speed $w_s$ for the particle species used in our analysis (250–500 μm). For this, we use the general formula for particle settling speeds[48] with coefficients for settling tephra particles determined using tank experiments by[40] (see Methods). A representative settling speed for this group is $w_s \approx 3 \pm 1 \ \text{cm s}^{-1}$.

Using the inferred dispersal scale of $L = 4.9 \pm 0.4$ km and the representative settling speed, $w_s \approx 3 \pm 1 \ \text{cm s}^{-1}$, the inversion formula of Eq. (2), $Q_{umb} = w_s L^2$, predicts that the volumetric rate of growth of the umbrella was $Q_{umb} \approx (7.6 \pm 3.6) \times 10^5 \ \text{m}^3 \ \text{s}^{-1} \approx 2.8 \pm 1.3 \ \text{km}^3 \ \text{hour}^{-1}$. In turn, the implied rate of heat transfer at the hydrothermal source predicted by Eq. (3) is $\Phi \approx (5.5 \pm 3.3) \times 10^{15} \ \text{J hour}^{-1}$ or $1.5 \pm 0.9$ TW (assuming a point source or a fissure length <1 km, which is most appropriate based on analysis of the NESCA bathymetry[49]; a source fissure longer than 1 km results in a slight decrease in the lower bound for the predicted energy flux; see Fig. 5). In evaluating Eq. (3), we used a value for the Brunt-Väisälä frequency $N \approx 10^{-3} \ \text{s}^{-1}$ derived for the seafloor near the NESCA site using the dataset of [50] (see Methods). These inferences of $Q$ and $\Phi$ represent the first constraints on rates of umbrella growth and heat energy input for a submarine eruption, as well as the first derived from buoyancy-driven tephra-dispersal dynamics in a plume umbrella.

It is important to note that since the energy flux at the plume source is constrained via the dispersal data, it characterizes the rate of energy transfer occurring during the period of hydrothermal discharge that was coincident with the generation of tephra. Estimating the total energy content of this tephra-bearing plume therefore requires some knowledge of the likely duration of the pyroclastic phase of the eruption. Observations from multiple submarine volcanic locations show that deep-sea tephra deposits of the kind sampled at NESCA are consistently associated with lava morphologies produced during high-effusion rate eruptions (i.e., sheet-flows)[15]. Conversely, low effusion rate pillow-lava forming eruptions do not appear to produce significant amounts of fragmental material[15]. Constructing a detailed facies architecture of the NESCA lava and resolving a precise eruption chronology will require analysis of high resolution mapping data (e.g.[51]), however it is possible to use existing observations to estimate the likely duration of the pyroclast-producing phase of the eruption.

The NESCA lava exhibits both pillow-lava and sheet-flow morphologies[38], implying a range of effusion rates over the course of the eruption (most likely ~$10^1$–$10^2 \ \text{m}^3 \ \text{s}^{-1}$ [51,52]). The higher effusion rate (sheet-flow forming) phase appears to have occurred first[38], as is common for basaltic eruptions[51,53]. Based on visual seafloor observations, these flows are estimated to account for a third of the erupted material[38], corresponding to an erupted volume of ~$1.5 \times 10^7 \ \text{m}^3$ (based on the total erupted volume given by[15]). Using a typical range of volumetric discharge rates for submarine sheet-flows of ~200–500 $\text{m}^3 \ \text{s}^{-1}$ [51], we estimate the duration of this activity, and therefore the probable timescale over which the tephra bearing plume was formed, to be $\tau \sim 10$–20 h.

Using the volume flux $Q_{umb}$ and heat transfer rate $\Phi$ constrained by our model, this estimate of eruptive duration would produce a plume with a volume of ~15–80 km³ containing ~$2$–$20 \times 10^{16}$ J of heat. These both lie directly within the ranges of total volume and total energy contents from observations of megaplumes shown as green bars in Fig. 4. We therefore infer that a megaplume was generated synchronously with the eruption

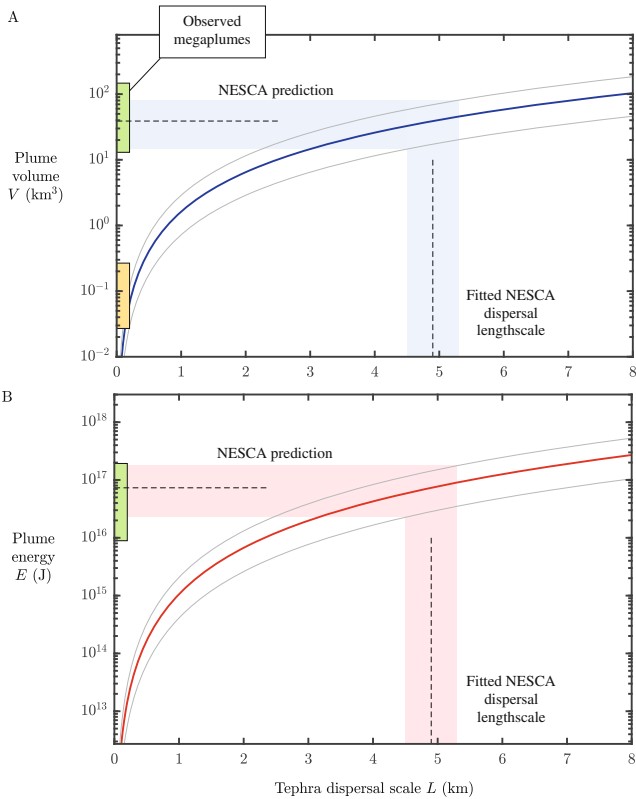

**Fig. 4 The inversion results for plume umbrella volume and total energy and comparison with observed event plumes.** (**A**) The total volume and (**B**) total energy predicted by our model as functions of the observed tephra dispersal lengthscale (L), defined as the decay scale of the Gaussian model of Eq. (1) (and equal to the distance from the center encompassing ~93% of the mass of tephra dispersed by the umbrella of the particle group under consideration). The value used for the Brunt-Väisälä frequency is $N = 10^{-3}$ $s^{-1}$ and the eruptive timescale used to convert our energy flux into a total energy is taken here as the representative value $\tau = 15$ h (as derived in our results section, where a range of 10–20 h is inferred based on the lava volume and morphology). The thick curves are evaluated for a settling speed of $w_s = 3$ cm $s^{-1}$, for the particle range 250–500 μm used in the inversion. The other curves (gray) represent the minimal and maximal inferences that would apply for settling speeds of $w_s = 2$ cm $s^{-1}$ and duration $\tau = 10$ h, and $w_s = 4$ cm $s^{-1}$ and duration $\tau = 20$ h, covering ranges of uncertainty in these parameters. Bands represent the inferred values based on the fitted dispersal length scale $L = 4.9 \pm 0.4$ km determined by our fitting to the observed data for the Northern Escanaba (NESCA) eruption (Fig. 2). The range of volumes and heat energies of observed megaplumes (volume ≥10 km³)[5] are indicated by the green bars along the vertical axes, showing consistency with both of our predictions. This indicates that a megaplume was produced during the NESCA eruption. The volume range of the considerably smaller group of event plumes observed at the Lau Basin[6] are indicated by the orange bar in (**A**), potentially forming a distinct category of event plume.

of lava and dispersed tephra during its radial propagation along a neutral buoyancy level. Although the period of plume formation is somewhat uncertain, maintaining the energy flux necessary to disperse the tephra ( ~1–2 TW) over any reasonable eruptive period (hours to days) would produce a plume with physical characteristics within the range of observed megaplumes (Fig. 4). Our results thus support a direct causal link between active lava effusion, megaplume generation and the km-scale dispersal of tephra in the oceans.

**Plausibility of a purely volcanic origin for megaplumes.** A key controversy surrounding megaplumes is whether the energy that drives plume formation is supplied directly from cooling magma, either from seafloor lavas[14,15] or in a subsurface dyke[16], or alternatively is predominately sourced from the rapid evacuation of intracrustal hydrothermal systems[2,54,55]. The temporal-spatial correlation between several observed megaplumes and active/recent seafloor volcanism[4,5,9,11] provides circumstantial evidence for a direct causal relationship between megaplume generation and eruptive activity. Our results support this association as they directly link megaplume formation and magma extrusion in both time and space. Debates on the feasibility of a volcanic/magmatic source have generally focused on the total energy contents of megaplumes as well as the origin(s) of the physico-chemical characteristics of megaplume fluids, such as ³He/Heat ratios and Fe and Mn concentrations[6,12,15,56–58]. However, since neither the actual timescale of plume generation nor the precise temporal relationship with magma extrusion is known, these debates remain inconclusive. Since our model constrains the actual energy flux released during the eruption, independent from both the overall timescale of plume formation and total energy budget, we are able to advance this debate by taking the more direct approach of evaluating whether the rate of heat transfer expected from cooling lava and/or exsolved magmatic volatiles can create the necessary buoyancy flux at the plume origin.

Upon eruption, the outer layer of submarine lava is rapidly quenched to form a solid insulating crust. The initial quenching of this outer layer is relatively fast, with a 1 mm thick crust forming in <1 s[59]. The temperature at the lava-water interface, and therefore the heat flux to the water, decreases as this conductive boundary layer thickens until either the flow is completely solidified or steady conditions are attained (dependent on the flow thickness, the magma supply rate, etc.). For flow thicknesses of ≥2–3 m, theoretical calculations based on a mathematical model of heat conduction predict that the first few days of cooling are characterized by a waning heat flux at the lava-water interface within the range of $10^3$–$10^4$ W m$^{-2}$ [60]. In order to create the observed tephra dispersal at NESCA via lava heating alone, the results of our inversion indicate that the integrated energy flux from the surface of the lava during megaplume generation must have been at least ~1 TW (and possibly up to 2 TW). A heat flux within the first day of cooling of order $10^4$ W m$^{-2}$ [60] would require ~100 km² of lava, almost a factor of seven higher than the total area of the NESCA eruption (estimated to be 15 km² [15]).

Heat loss from fragmented magma (pyroclasts) is more efficient than the cooling at the surface of a lava flow assumed in the estimate above. However, for typical MOR eruptions the mass fraction of fragmental material appears to be so low (for example at NESCA it is estimated to be <1 wt %[15]) that this cannot provide a significant source of heat. Given the typical areal extent of most submarine lava flows (<10 km²), we can anticipate that heat transfer from erupted magma is unlikely to be the dominant mechanism of megaplume generation.

An alternative proposition for the direct heating of water by magma is that megaplume fluids are created via heating of pore fluids along the edges of a dyke[16]. However, this process also requires a prolonged period of heat transfer (>10 days, even at high crustal permeabilities[16]) in order to create sufficient volumes of hydrothermal fluid. Therefore, it is similarly unable to provide the heat transfer rate necessary to create the required buoyancy flux for tephra dispersal.

A further possibility for a direct volcanic origin for mega-plumes is heating of seawater by a separate fluid phase composed of exsolved magmatic volatile species[15], similar to the formation of subaerial eruption plumes. Although volatile exsolution is

relatively limited during seafloor eruptions (and MOR magmas are themselves typically volatile poor), some bubble growth is probably necessary to provide the buoyancy for eruptive ascent[31] and a $CO_2$ rich fluid phase is likely to be present in the erupting magma. The most $CO_2$ rich MORB magmas probably have initial dissolved $CO_2$ contents of ~1.5 wt%[61], which for a NESCA sized eruption would transport around ~$10^{15}$ J of heat (assuming an initial temperature of 1200 °C, complete $CO_2$ exsolution, and a specific heat capacity of $c_p = 1.3$ kJ kg$^{-1}$ K$^{-1}$ [62]), one to two orders of magnitude below that required to form a megaplume. However, if explosive eruptive styles do occur on the seafloor then it is likely they are driven by accumulations of $CO_2$ rich fluid, possibly from a foam layer at the roof of the subsurface magma reservoir[15,19,31,32,63]. An energy flux of ~1 TW would require a $CO_2$ flux of ~$10^6$ kg s$^{-1}$ (equivalent to ~$10^4$ m$^3$ s$^{-1}$ at the depth of the NESCA vent). While a $CO_2$ output of this magnitude may be viable for short periods, it seems unlikely that it could be sustained over the period of hours required for megaplume formation. Nevertheless, if megaplume generation is associated, at least partly, with volcanic energy release, then heat loss from exsolved $CO_2$ rich bubbles is a more feasible energy source than cooling magma and would be consistent with the proposed mechanism for explosive MOR eruptions[15,32]. A direct input of exsolved magmatic volatiles may also provide a better explanation for some of the distinctive chemical features of megaplumes, such as their $^3$He/heat ratios, compared to magma-water interaction[15,56,57].

Based on these estimates, we anticipate that the rapidity of megaplume formation at NESCA, and probably elsewhere[6], likely requires an additional syn-eruptive energy source. The most obvious mechanism for generating this energy flux is the rapid evacuation of pre-existing intra-crustal hydrothermal fluids triggered by the mechanical effects of dyke intrusion[2,54,55]. A crustal origin for at least a portion of megaplume fluids is consistent with the presence of (crustally-derived) thermophilic microbes observed in plume fluids at the Gorda Ridge in 1996[64]. The high rates of energy transfer required for tephra dispersal suggests that these fluids may provide the dominant contribution to megaplume heat contents.

Seafloor eruptions that do not involve the concurrent release of significant volumes of crustal fluid, perhaps due to a lack of available hydrothermal fluids or magma ascent conditions that do not promote fluid discharge, should therefore produce substantially smaller plumes. The size of such plumes would be limited by the more restricted/short-lived energy transfer available from volcanogenic heating only. This may explain the formation of a series of event plumes of considerably smaller volumes $V < 0.5$ km$^3$ (indicated by the orange bar in Fig. 4A) detected after an eruption in 2008 in the Lau Basin[6]. It is notable in this regard that megaplumes have so far only been observed, or suspected, above volcanoes in extensional tectonic settings (i.e., MORs and back-arc ridges; Fig. 1), where seafloor hydrothermal systems are commonplace and the extensional tectonic regime promotes hydrothermal circulation, which may be significantly enhanced by dyke intrusions[55]. Although future observations may yet document megaplumes associated with other submarine volcanic environments, it could be expected that syn-eruptive plumes in non-rift settings are typically less energetic. This inference could be tested by applying our inversion method to plume dispersed pyroclasts from other submarine volcanic settings, such as submerged volcanic arcs (e.g.[25]).

## Discussion
The release of hot intracrustal fluid triggered by magma intrusion does not, in principle, necessitate an eruption. However it is notable that observed megaplumes are commonly associated with

events involving lava extrusion[58] and appear to form directly above freshly emplaced lava flows[9]. It has also been noted that, on occasions when seafloor seismic events have not culminated in eruptions, no anomalous hydrothermal activity has been detected[58] (although the detection and observation of active seafloor eruptions remains extremely challenging). As demonstrated here, in the absence of strong unidirectional currents, achieving transport distances in excess of 1 km for even relatively small pyroclasts (<500 μm) requires a significant time-averaged energy flux close to the eruptive source of around 1–2 TW. Maintaining this heat flux for a period of hours would equate to a total energy release consistent with the observed heat contents of megaplumes ($10^{16}$–$10^{17}$ J), shown in Fig. 4B. The apparent ubiquity of widely dispersed submarine tephras (Fig. 1), and the aforementioned correlation between lava extrusion and megaplume detection, both indicate that syn-eruptive energy transfers of a magnitude comparable to that predicted for the NESCA eruption are an intrinsic characteristic of many volcanic events occurring in the deep oceans. If the energy transfer driving megaplume creation is primarily associated with rapid evacuation of intracrustal fluid reservoirs rather than magmatic heating then we anticipate that the processes that instigate this fluid release (i.e., dyke intrusion into the uppermost crust) will also generally produce lava extrusion. It may also be the case that the initial phase of seafloor eruptions are often characterized by enhanced rates of energy transfer via exsolved volatiles due the involvement of $CO_2$ rich foams.

Finally, we note that our methodology of inversion based on tephra-transport within the umbrella of a hydrothermal plume can be anticipated to apply to general situations where the ambient is sufficiently quiescent that it is unable to compete significantly with the buoyancy-driven radial flow during particle dispersal, and the particle concentration is small enough such as not to significantly affect the fluid dynamics. Tephra transport from submarine eruptions can occur via other mechanisms, such as density currents generated by plume collapse[25,35] or the advection of settling particles by ocean currents[40], however these will not produce radially dispersed deposits over length-scales of several km with Gaussian thinning trends (as are clearly apparent in the NESCA data; Fig. 2). While our analysis reveals primarily buoyancy-driven radial ash transport within a megaplume, our method is equally applicable to tephra transported by any eruptive plume, regardless of size or total energy content, and could be generalised to include the effects of ambient diffusivity and/or crossflow. In situations where background crossflow contributes, our methodology could be adapted by comparing observed tephra depositions alongside a fluid-mechanical model of the umbrella suitably generalised to incorporate background flow and fitting for the background flow rate (if unknown) and flux feeding the umbrella simultaneously. The signature of the flux feeding the umbrella and, in turn, the heat flux at the seafloor origin, will manifest in the lateral buoyancy-driven expansion of the flow perpendicular to the direction of the crossflow.

In this work we have demonstrated how the horizontal buoyancy-driven transport of tephra in a plume umbrella (in this case from a submarine deposit) can be used to constrain the energy discharge rates associated with volcanic eruptions. Our method presents a novel approach to invert tephra dispersal data for eruptive energetics. By applying this model to the submarine tephra deposit from the NESCA eruption, we have shown that km-scale tephra dispersal in the deep ocean can be explained by buoyancy-driven transport in a syn-eruptive megaplume and our results conclusively link megaplume generation with the tephra-generating phase of this eruption. The similarity between the NESCA tephra deposit and many other deep marine tephras (wrt. particle size, morphology, dispersal range etc.) suggests that this is

a common occurrence during submarine eruptions at ocean ridges. Although our results demonstrate a clear temporal-spatial correlation between megaplume formation and seafloor eruptions, the primary energy source for megaplume creation seems unlikely to come directly from the erupted magma. While some portion of the heat transfer that drives megaplume creation must be derived from the concurrent volcanic eruption, via cooling magma and, perhaps more significantly, exsolved $CO_2$, it seems likely that the high rates of energy release required to transport submarine tephras are associated with the rapid evacuation of hydrothermal reservoirs[2,6,7], probably in response to dyke intrusion into the uppermost crust[55]. This inference can potentially be tested by future in situ observations of syn-eruptive hydrothermal processes[65] and continued sampling and chemical analysis of megaplume fluids. Application of our inversion method to paleo-tephra deposits recovered from marine sediment cores[29,30] could, in principle, provide new inroads toward constraining the long-term ($\gtrsim 10$ ka) time-averages of the flux of mass and heat from the crust to the oceans associated with seafloor volcanic events.

## Methods

**Plume model and horizontal buoyancy-driven dispersal.** Our model of tephra transport is based on advection within the neutrally buoyant component of a turbulent hydrothermal plume generated by a heat input at the seafloor. The stem of the plume is modeled as a turbulent column of hot water that propagates vertically within the ambient density stratification of the ocean. The turbulent flow in the stem of the plume will entrain seawater, causing it to cool and eventually rise to a height where it is heavier than the seawater. Following an inertial overshoot, the flow will settle along a neutral level, forming the umbrella. This second regime forms a turbulent, horizontally propagating flow known as an intrusion or neutrally buoyant gravity current e.g.[66]. The stem and umbrella are coupled by a condition of continuous volumetric flux $Q_{umb}$ between the top of the stem and the radial source of the umbrella at a neutral buoyancy level.

Particles entrained into the plume will propagate into the plume umbrella and sediment from its base at a rate proportional to particle concentration[42,43]:

$$\frac{\partial (ch)}{\partial t} + \frac{1}{r}\frac{\partial}{\partial r}(rchu) = -w_s c, \tag{5}$$

where $c(r,t)$ is the volume concentration of particles, $r$ is the horizontal distance from the plume center, $h(r,t)$ is the thickness of the umbrella layer, $u(r,t)$ is the thickness-averaged horizontal velocity of the flow, and $w_s$ is the settling speed of the particle species being considered. The right-hand side represents the rate of particle fallout, which is modeled as proportional to the concentration and the settling speed[42,43,67,68]. Numerical analysis of the fully time-dependent gravity-current equations[66] shows that the current extends along a near-steady envelope. Consequently, the condition of uniform flux $2\pi rhu \equiv Q_{umb}$ applies to excellent approximation in Eq. (5) during the growth of the umbrella. Using this expression to substitute for $hu$ in Eq. (5) and integrating the resulting equation for $c$, one obtains the Gaussian prediction for the spatial distribution of the mass per unit area of ash deposited per unit time:

$$\Omega(r) \equiv \rho_b c = \Omega_0 e^{-\pi(r/L)^2}, \tag{6}$$

where $\rho_b$ is the density of the basaltic glass, $L = (Q_{umb}/w_s)^{1/2}$ is referred to herein as the umbrella dispersal lengthscale, and $\Omega_0$ is the integration constant. The result of Eq. (6) applies downstream of the maximum radius of the plume stem, $r_0$. Assuming that the settling particles are not significantly advected from their fallout position (consistent with the assumption of an approximately quiescent ambient), Eq. (6) provides the deposition profile of tephra along the seafloor. The constants $\Omega_0$ and $L$ form the only two parameters defining the dispersal pattern (Eq. (6)) and describe two independent degrees of freedom. The constant $\Omega_0$ represents the accumulation of the particles, and is related to the rate of particle generation and source duration (a larger eruptive duration will accumulate a larger mass per unit area but the dispersal profile will retain the same shape). The parameter $L = (Q_{umb}/w_s)^{1/2}$ provides the radial rate of decay of the deposition profile, and independently represents the structure of the distribution. It contains information of the rate of fluid input into the umbrella $Q_{umb}$ and the settling speed $w_s$ representing the particle size under consideration. A larger flux feeding the umbrella $Q_{umb}$ or a smaller particle size (i.e., a smaller settling speed $w_s$) lead to larger dispersal length scales $L$. The axisymmetric dispersal pattern predicted by Eq. (6) decays monotonically with a smooth tail in all directions from the plume center. With the dispersal lengthscale $L$ inferred by fitting Eq. (6) to an observed tephra deposition profile, the volumetric flux feeding the umbrella can then be inferred using the formula

$$Q_{umb} = w_s L^2. \tag{7}$$

Since $L$ is independent of both the duration of the eruption and the rate of input of particles (either of which will only accumulate $\Omega_0$), $L$ independently constrains the volumetric flux sourcing the umbrella via Eq. (7). Hence, even a small input of particles can, in principle, be sufficient to conduct an inversion for $Q_{umb}$ using Eq. (7). To invert for $Q_{umb}$, it remains only to estimate the settling speed of the particle under consideration $w_s$ and the dispersal length $L$ for the observed deposition distribution.

In the analysis above, it has been assumed that the particle species is represented by a single representative settling speed $w_s$. Volcanic tephra will naturally involve a polydisperse distribution. As discussed in Supplementary Note 2, the deposition field resulting from a polydisperse distribution can, under our assumption of a dilute suspension (see below), be represented as an integral superposition of Gaussians of the form (6), in which the integrand is weighted by a mass distribution function representing the concentration of particles as a function of particle size $d$. To apply our inversion, we choose a group of particles in the size range $[d - \delta d/2, d + \delta d/2]$ and assign a representative settling speed $w_s$ corresponding to the central value $d$. The data of[15] was partitioned using sieving into four categories, and we use the particle range 250–500 μm for our analysis. A representative range of settling speeds for this range of sizes is given by $w_s = 3 \pm 1$ cm s$^{-1}$ (see below).

The second step of our inversion predicts the heat energy $\Phi$ inputted into the hydrothermal plume at its base necessary to produce the volumetric flux $Q_{umb}$ feeding the umbrella at the top of the stem. The transfer of heat energy, either from inputted hot fluid or heating by lava, produces plume fluid through the process of entrainment of ambient seawater caused by the turbulent upwelling of the plume[3]. If the seafloor heat input is localized along a fissure, then a linear heat input is more appropriate. If the length of the source is sufficiently long (a prediction for how long will be determined below), a planar model of the plume stem may be most appropriate[3,17]. If the fissure length $l$ is sufficiently short (and/or the intensity of the eruptive source sufficiently centralized), then the finitude of the source (edge effects) will invalidate the assumption of planarity. A point-source model will then 'take over' as being the more applicable. We develop models of both of these limiting endmember cases, and compare their predictions together. It should be noted that, irrespective of the geometry of the stem, it is clear from the characteristics of the NESCA deposition profile (Fig. 2), particularly its conformity with Gaussian axisymmetric dispersal, that the umbrella was primarily radially spreading (as opposed to primarily one-dimensional flow perpendicular to a fissure strike, for example, which would create an exponential, as opposed to Gaussian, decay in only the two horizontal directions perpendicular to the fissure). Predominately radial flow of the umbrella at the large scale is possible despite potential sourcing by a planar seafloor origin, either because the flow of the umbrella will lose information of the details of its source geometry beyond a characteristic distance owing to lateral buoyancy-driven spreading, and/or because the stem will approach an axisymmetric plume during ascent (for sufficiently short fissure lengths). In either case, we propose a "hybrid" model in which either a predominantly axisymmetric or planar seafloor source feeds the approximately radially spreading umbrella.

In the limit of an axisymmetric stem, we apply the model of a vertically flowing plume given by[69], as specified by

$$\frac{dQ}{dz} = 2\sqrt{\pi}\varepsilon M^{1/2}, \quad \frac{dM}{dz} = \frac{FQ}{M}, \quad \frac{dF}{dz} = -N^2 Q, \tag{8}$$

where $z$ is the vertical coordinate with respect to the seafloor, $N$ is the Brunt-Väisälä frequency, $\varepsilon \approx 0.1$ is the entrainment coefficient, and $Q(z)$, $M(z)$ and $F(z)$ are the volume, momentum and buoyancy fluxes, respectively. It should be noted that the model above describes only the predominantly vertically flowing stem of the plume, and does not apply in the neutrally buoyant umbrella for which the earlier model of Eq. (5) applies. Let $F_0$ denote the buoyancy flux introduced at the base of the plume. By considering the intrinsic scalings of Eq. (8) and the scaling implied by the source condition, $F \sim F_0$, we determine the unique intrinsic flux scale in the system as $Q_* = (\varepsilon^2 F_0^3 / N^5)^{1/4}$. Solving Eq. (8) numerically using a Runge-Kutta integrator subject to the heat-source condition $F = F_0$ and $Q = M = 0$ at $z = 0$, we determine the prefactor to this intrinsic scale giving the explicit formula for the flux at the top of the plume: $Q_{umb} = 3.52 \, Q_* = 3.52(\varepsilon^2 F_0^3 / N^5)^{1/4}$. This provides the desired relationship between the input of buoyancy at the seafloor $F_0$ and the flux feeding the umbrella $Q_{umb}$. On rearranging this expression for the source buoyancy flux $F_0$, we obtain

$$F_0 = 0.187 \left( \frac{N^5 Q_{umb}^4}{\varepsilon^2} \right)^{1/3}, \tag{9}$$

which provides the buoyancy flux needed to generate the flux $Q_{umb}$ feeding the umbrella at the top of the plume stem. Thus, once $Q_{umb}$ has been determined from the first stage of our inversion, the expression above represents the second stage to infer the source buoyancy flux (which, as will be shown below, can then be converted into an energy flux).

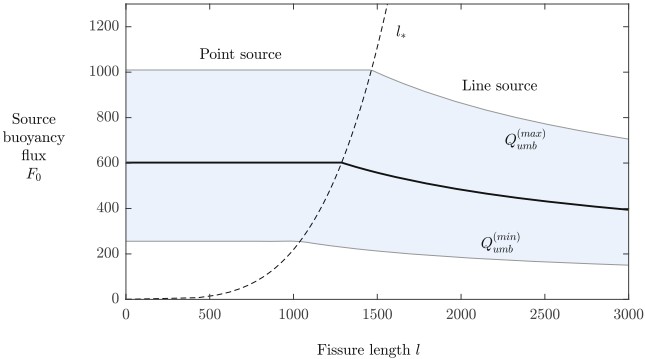

**Fig. 5 Inferred buoyancy flux as a function of fissure length, illustrating the transition from a point-source model for the plume stem to a line-source model.** The blue shading represents the range of inferred values of the buoyancy flux at the plume source $F_0$ given the full range of umbrella fluxes, $Q_{umb}^{(min)} < Q_{umb} < Q_{umb}^{(max)}$, predicted by Eq. (4), as a function of the length of the source $l$. For a point source, or sufficiently small fissure lengths ($l \lesssim l_*$), the details of the source are unimportant to good approximation and the predictions conform to those of a point-source model, as given by Eq. (9). For sufficiently long sources ($l \gtrsim l_*$), a model assuming a planar source becomes more applicable in accordance with the prediction of Eq. (11). The lengthscale $l_*$, given by Eq. (12) and indicated by a dashed curve, represents the fissure length on which the predictions of the two theories are equivalent.

Under the assumption of a planar stem applicable to sufficiently long fissures, we consider the two-dimensional analog of the model of Eq. (8) specified by:

$$\frac{dq}{dz} = -2\varepsilon \frac{m}{q}, \quad \frac{dm}{dz} = \frac{fq}{m}, \quad \frac{df}{dz} = -N^2 q, \qquad (10)$$

where $q$, $m$, and $f$ represent the volume flux, momentum flux and buoyancy flux per unit width of the fissure, and $\varepsilon$ is the entrainment coefficient[3]. We assume that the plume is sourced by a buoyancy flux per unit length $f_0 = F_0/l$, where $F_0$ is the total source buoyancy flux and $l$ is the fissure length. In this model, it is assumed that the fissure is long enough that edge effects from the ends of the fissure are negligible (a condition for this assumption to apply to good approximation will be derived below). A scaling analysis of the equations above determines the fluid flux per unit length at the top of the stem: $q_{umb} = 2.11 \, (\varepsilon f_0^2)^{1/3}/N$, where we have used a numerical solution to determine the dimensionless prefactor. Recasting this expression in terms of the total source buoyancy flux and the umbrella volume flux using $f_0 = F_0/l$ and $q_{umb} = Q_{umb}/l$, and rearranging for $F_0$, we obtain

$$F_0 = 0.326 \left( \frac{N^3 Q_{umb}^3}{\varepsilon l} \right)^{1/2}. \qquad (11)$$

This result represents the planar analog of Eq. (9). For a given umbrella flux $Q_{umb}$, the buoyancy flux $F_0$ predicted by Eq. (11) decreases with fissure length $l$ because a longer source produces a larger surface area along the sides of the plume, and hence more efficient entrainment.

If a detailed numerical or experimental study of plumes generated by finite line sources were conducted, then the results of Eqs. (9) and (11) above can be expected to provide $l \to 0$ and $l \to \infty$ asymptotes, respectively. Since the planar theory will breakdown for sufficiently small fissure lengths, the relevant theoretical prediction will switch to the axisymmetric endmember (the axisymmetric theory represents a theoretical upper bound as $l \to 0$). The predictions of the models of Eqs. (9) and (11) are equivalent at a fissure length of

$$l_* = 3.0 \left( \frac{\varepsilon Q_{umb}}{N} \right)^{1/3}, \qquad (12)$$

which we propose characterizes the fissure length on which the transition between the axisymmetric and planar theories occurs (it contains the unique length scale that can be formed from $Q_{umb}$ and $N$ alone). The two predictions are illustrated together in Fig. 5, where we have assumed the values of $N = 10^{-3} \, \text{s}^{-1}$ and $Q_{umb} = (7.6 \pm 3.6) \times 10^5 \, \text{m}^3 \, \text{s}^{-1}$, with a dashed curve showing the anticipated transition between the two theories predicted by Eq. (12).

Finally, we convert the inferred buoyancy flux at the source predicted by either the axisymmetric or the planar theories derived above into a flux of heat energy using the expression $\Phi = k F_0$, where $k \equiv \rho c/\alpha g$ is the conversion factor between buoyancy flux and heat flux, $\rho$ is the density of seawater, $c$ is the specific heat capacity, $\alpha$ is the coefficient of thermal expansion, and $g$ is the acceleration due to

gravity. The two expressions for $F_0$ above yield

$$\Phi \approx \begin{cases} 0.187 \, k \left( \frac{N^5 Q_{umb}^4}{\varepsilon^2} \right)^{1/3} & \text{(axisymmetric, } l \lesssim l_*\text{),} \\ 0.326 \, k \left( \frac{N^3 Q_{umb}^3}{\varepsilon l} \right)^{1/2} & \text{(planar, } l \gtrsim l_*\text{).} \end{cases} \qquad (13)$$

A typical value for the conversion factor is $k \approx 2.1 \, \text{GW m}^{-4} \, \text{s}^3$, using values of $\rho \approx 1027 \, \text{kg m}^{-3}$, $c \approx 4200 \, \text{J kg}^{-1} \, \text{K}^{-1}$, $\alpha \approx 2.1 \times 10^{-4} \, \text{K}^{-1}$ and $g \approx 9.8 \, \text{m s}^{-2}$. Having determined the volumetric flux into the umbrella $Q_{umb}$ using the first stage of our inversion using Eq. (7), the result of Eq. (13) provides the flux of heat energy introduced into the plume system at its seafloor origin necessary to produce this flux.

Given the interval of fluxes feeding the umbrella predicted from the first stage of our inversion ($4.0 \times 10^5 < Q_{umb} < 11.2 \times 10^5 \, \text{m}^3 \, \text{s}^{-1}$), Eq. (13) yields a prediction for the source buoyancy flux $F_0$ shown in Fig. 5. In this figure, the dashed curve represents the transitional length scale $l_*$ separating the two theoretical predictions resulting from variation of $Q_{umb}$ given by Eq. (12). This predicts that a longer fissure results in a smaller inferred flux of heat input $F_0$ (for fixed $Q_{umb}$), consistent with entrainment being more efficient for longer fissure lengths $l$. In view of the bathymetry of the NESCA site indicating that the eruption had a central origin[49] (see Fig. 2), we adopt the range of inferred values of $\Phi$ predicted by the axisymmetric model in our analysis.

**Tephra transport dynamics in the stem and the conditions for dominant umbrella dispersal.** This section develops theoretical conditions for umbrella dispersal to occur. This is done first by considering a necessary condition for significant umbrella dispersal, namely, that the dispersal distance predicted by the dynamics of the umbrella is considerably larger than the radius of the plume stem. Second, we develop a theory for the transport of particles in the stem of the plume. By using this to determine the proportion of particles at the top of the stem, we determine a condition for a significant proportion of particles to reach the umbrella. The two conditions derived are found to involve the same intrinsic dimensionless parameter: $\Gamma = w_s/(NF_0)^{1/4}$.

*Condition for umbrella dispersal.* A necessary condition for significant dispersal by the umbrella is that the umbrella dispersal scale $L$ is larger than the maximum radius of the plume stem, $r_0$. As a metric to assess the satisfaction of this condition, we define the umbrella dispersal parameter:

$$\Lambda \equiv L/r_0, \qquad (14)$$

representing the ratio of the umbrella dispersal length scale $L$ to the maximum stem radius $r_0$. If $\Lambda \gg 1$, then the umbrella-driven dispersal considerably exceeds the maximum distance that can be dispersed by the stem, which is consistent with the former being the dominant process. Conversely, if $\Lambda \ll 1$, dispersal will be limited to the region below the plume stem. To determine $\Lambda$ in terms of the intrinsic parameters specifying the plume-particle system ($F_0$, $N$, and $w$), we substitute the relationships for $L$ and $r_0$ given by our theoretical model presented above. First, we recall from Eq. (7) that $L = (Q_{umb}/w_s)^{1/2}$ and from the text below Eq. (8) that $Q_{umb} = 3.52(\varepsilon^2 F_0^3/N^5)^{1/4}$ (assuming the axisymmetric form of the model here for now). To estimate $r_0$, we use the expressions for the cross-sectional area and radius of the stem given by $A = Q^2/M$ and $r(z) = \sqrt{A/\pi}$ in accordance with the top-hat form of the model of[69]. From scaling analysis and a numerical integration of Eq. (8), we can estimate the maximum radius of the stem to be $r_0 \approx \varepsilon (F_0/N^3)^{1/2}$. Substituting these expressions into Eq. (14), we determine the umbrella dispersal parameter given by Eq. (14) as a function of the intrinsic parameters:

$$\Lambda \approx 3 \left( \frac{F_0 N}{w_s^4} \right)^{\frac{1}{8}} \equiv 3 \, \Gamma^{-1/2}. \qquad (15)$$

The result reveals an intrinsic dimensionless parameter grouping $\Gamma = w_s/(F_0 N)^{1/4}$ controlling the relative significance of umbrella dispersal. Appreciable umbrella dispersal ($\Lambda \gtrsim 1$) will occur for $\Gamma \lesssim 10$. Thus, larger plume buoyancy fluxes (larger $F_0$) and stronger stratifications (larger $N$) result in relatively more dispersal through the umbrella. For characteristic values of $F_0 \approx 600 \, \text{m}^4 \, \text{s}^{-3}$, $N \approx 10^{-3} \, \text{s}^{-1}$ and $w_s \approx 3 \, \text{cm s}^{-1}$ arising in our analysis of the 250–500 μm particle range in the NESCA eruption, we obtain $\Gamma \approx 0.034$, and hence $\Lambda \approx 16$, consistent with significant distances of dispersal via the umbrella.

*Particle transport.* We develop here also a model to predict the proportion of particles reaching the umbrella. To do this, we begin by generalizing a theory of Ernst et al.[70] describing the particle concentration in a plume stem to the case of a general plume and stratification. Following[70], we describe the particle mass concentration, $c(z, t)$, using the particle conservation equation:

$$\frac{\partial}{\partial t}(Ac) + \frac{\partial}{\partial z}(Qc) = -w_s(2\pi r)c, \qquad (16)$$

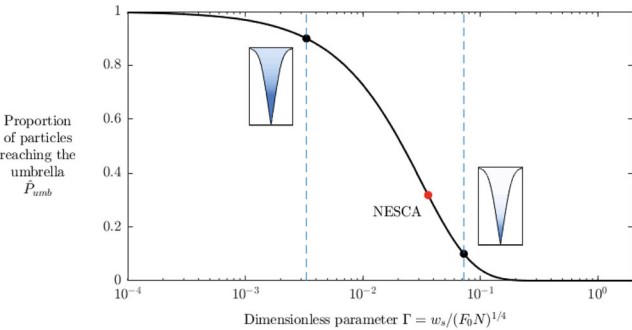

**Fig. 6 Theoretical model for the proportion of particles reaching the umbrella of the eruptive plume.** The prediction for the proportion of particles reaching the umbrella given by Eq. (20) of the Methods as a function of the key dimensionless number $\Gamma = w_s/(F_0 N)^{1/4}$. The result shows that over 90% of particles reach the umbrella if $\Gamma < 0.003$ and under 10% reach the umbrella if $\Gamma > 0.07$. Insets illustrate the concentration field of the plume $c(z) = \hat{P}(z)/\hat{Q}(z)$ given by Eq. (21), with darker blue shading representing a higher concentration. The red filled circle represents the value $\Gamma \approx 0.034$ predicted for the Northern Escanaba (NESCA) eruption for the particle range 250–500 μm using our inferred values of $F_0 \approx 600$, $w_s = 3$ cm s$^{-1}$, for which ~32% of particles in this range are predicted to reach the umbrella.

where $A(z)$ is the horizontal cross-sectional area of the plume, $r(z)$ is the plume radius, $Q(z)$ is the volume flux of the plume, as determined by Eq. (8), and the right-hand side represents the rate of particle fallout. In essence, Eq. (16) is the analog of Eq. (5) for the stem. In these definitions and Eq. (16), the functions of $Q(z)$ and $M(z)$ are known a priori from the solution to the model of Eq. (8). Using the expressions for the area and radius, $A = Q^2/M$ and $r(z) = \sqrt{A/\pi}$, Eq. (16) can be rewritten as

$$\frac{\partial}{\partial t}\left(\frac{Q^2 c}{M}\right) + \frac{\partial}{\partial z}(Qc) = -2\sqrt{\pi} w_s M^{-1/2}(Qc). \quad (17)$$

In steady state, Eq. (17) forms an ordinary differential equation for the vertical mass flux of particles, $P(z) = Q(z)c(z)$, which we integrate to yield

$$P(z) = P_0 \exp\left(-2\sqrt{\pi} w_s \int_0^z M(\tilde{z})^{-1/2}\, d\tilde{z}\right), \quad (18)$$

where $P_0$ is a constant of integration representing the inputted mass flux of particles (at $z = 0$). The result provides the required vertical distribution of particles for any given plume solution (any $Q(z)$ and $M(z)$ determined a priori from Eq. (8)).

For the case of a pure plume in a constant stratification, we write the momentum flux and the vertical coordinate in terms of their non-dimensional counterparts by $M(z) = (F_0/N)\hat{M}(z)$ and $z = (F_0/\varepsilon^2 N^3)^{1/4}\hat{z}$, giving

$$\hat{P}(\hat{z}) = \exp\left(-\beta\Gamma\int_0^{\hat{z}} \hat{M}(\bar{z})^{-1/2}\, d\bar{z}\right), \quad (19)$$

where $\hat{P} = P/P_0$ is the normalized mass flux of particles ($\hat{P} = 1$ represents the input flux at $\hat{z} = 0$), $\beta = 2\sqrt{\pi/\varepsilon} \approx 11.2$ is a dimensionless prefactor and $\Gamma = w_s/(F_0 N)^{1/4}$ is the same intrinsic dimensionless number that appeared in Eq. (15) above. The result of Eq. (19) provides the flux of particles in the stem as a proportion of the flux inputted at the base.

Evaluating Eq. (19) at the top of the stem ($\hat{z} = 1.37$), we determine the proportion of particles reaching the umbrella:

$$\hat{P}_{umb} = \exp(-b\Gamma), \quad (20)$$

where $b \approx 33.7$, on using the fact that $\int_0^{1.37} \hat{M}(\bar{z})^{-1/2}\, d\bar{z} \approx 3.0$. As shown by the plot of the relationship between $\hat{P}$ and $\Gamma$ predicted by Eq. (20) in Fig. 6, the dimensionless number $\Gamma$ dials between situations where the vast majority of particles reach the umbrella ($\hat{P}_{umb} > 0.9$ for $\Gamma < 0.003$) to those in which the majority of particles fall from the umbrella ($\hat{P}_{umb} < 0.1$ for $\Gamma > 0.07$). Again, the dimensionless number $\Gamma$ has appeared in Eq. (20) as the key index for determining whether a given plume will disperse a particle species of settling speed $w_s$ primarily via the umbrella versus the stem. For $\Gamma \approx 0.034$, as predicted for the 250–500 μm particle range for the NESCA eruption (see above), the result of Eq. (20) indicates that 32% of the particles in the range of 250–500 μm will reach the umbrella. Thus, while significant dispersal of these particles reaching the umbrella will occur (in accordance with the condition derived in the subsection above), our theory predicts that a large proportion of the particles introduced at the source of the plume will fallout from the sides of the stem. The result indicates that a considerable

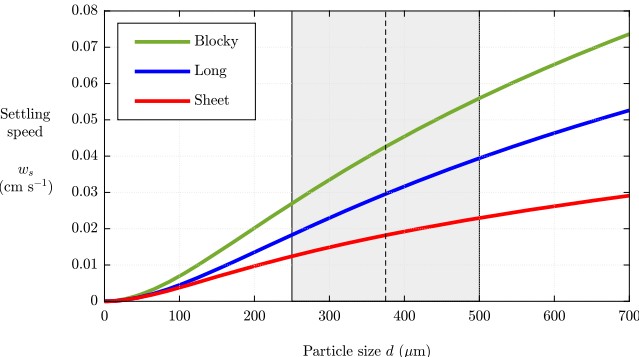

**Fig. 7 Settling speed as a function of pyroclastic particle size.** The settling speed $w_s$ of three different types of clast are plotted against the particle size $d$, as predicted by the formula of[48] with the coefficients for basaltic tephra determined from tank experiments by[40]. The relationships are used to determine the characteristic settling speeds of the particles in the 250–500 μm range, shown shaded, used to conduct our inversion for the volumetric flux of fluid fed into the umbrella using Eq. (2).

accumulation of tephra (as much as 70% of the total mass of tephra produced) was deposited near the vicinity of the source vent or fissure.

*Estimating tephra mass and concentration.* We can also apply the model above to estimate the particle concentration throughout the plume system. This is interesting in particular to assess the importance of particle dynamics on the evolution of the fluids. In our analysis, we have assumed a single-phase model for both the stem and umbrella, meaning that the presence of particles has a negligible effect on the fluid dynamics (producing a one-way coupling between the fluid dynamics and particle dynamics). This provides an excellent approximation if the mass concentration of particles $c$ is $<10^{-3}$. The mass concentration of particles (the particle mass per unit fluid mass) is defined by

$$c = \frac{P(z)}{\rho Q(z)}, \quad (21)$$

where $P(z)$ is the flux of particle mass per unit horizontal cross-section (as predicted by the theory above), $\rho Q(z)$ is the flux of fluid mass and $\rho \sim 10^3$ kg m$^{-3}$ is the density of water.

To estimate $P(z)$, we use the theoretical prediction of Eq. (20) above. For the purpose of checking the self-consistency of our assumption of a single-phase model, we will assume a maximal particle flux using the estimate of the total mass of tephra $M = 2 \times 10^7$ kg (predicted by[15]) and the shortest duration for the tephra-producing stage of the eruption estimated in our results section ($\tau = 10$ h), giving the mass flux of particles at the base of the stem as $P_0 \sim M/\tau \sim 500$ kg s$^{-1}$. Using the prediction for the particle mass distribution given by Eq. (18), and the prediction for the volume flux of plume fluid, $Q(z) = (\varepsilon^2 F_0^3/N^5)^{1/4}\hat{Q}(z)$, we use Eq. (21) to determine the mass concentration of particles as a function of height. The result shows that at least 99% of the stem has a particle mass concentration of $<10^{-3}$, and that the particle mass concentration at the top of the stem (and hence through the umbrella) is $<10^{-7}$. On this basis, single-phase models can be expected to apply to excellent approximation for describing the dispersal from the umbrella (which will only decrease below this value due to the effects of radial spreading and particle fallout) and throughout the large majority of the stem.

**Particle settling speeds.** In order to determine a representative settling speed for the category of particles sizes 250–500 μm used in our inversion, we utilized the relationship of[48] with coefficients for pyroclastic shapes determined using tank experiments by[40] (see Fig. 7). For the size range 250–500 μm used in our inversion (corresponding to the deposition shown in Fig. 2), the range $w_s = 3 \pm 1$ cm s$^{-1}$ is representative.

**Ocean density stratification.** The vertical profile of the Brunt-Väisälä frequency $N$ from the seafloor near the NESCA site is shown in panel A of Fig. 8, as measured by ARGO floats[50]. The blue profile corresponds to the data gathered at the location closest to the lava flow. The value of $N \approx 10^{-3}$ s$^{-1}$ is characteristic for the abyssal region. The surrounding 8 profiles are shown as black dots in panel B (on a horizontal grid with a spacing of 0.5 degrees). The profiles of $N$ for these additional 8 locations, shown as gray curves in panel A, show a consistent representative value of $N \approx 10^{-3}$ s$^{-1}$.

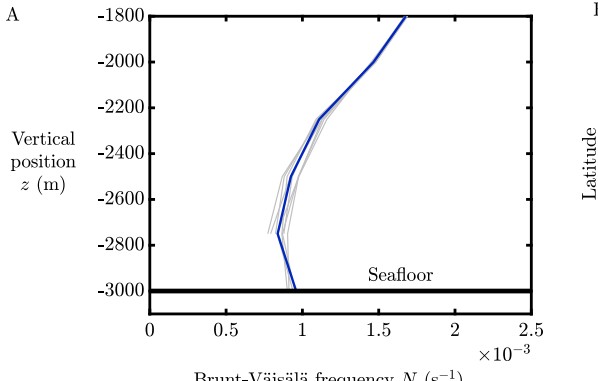

**Fig. 8 Profile of the Brunt-Väisälä frequency N as a function of height from the seafloor.** The profiles are provided by the dataset of[50]. (**A**) shows the profile of $N = [-(g/\rho_0)\partial\rho/\partial z]^{1/2}$ as a function of vertical position, where $g \approx 9.8$ m s$^{-2}$ and $\rho_0 \approx 1027$ kg m$^{-3}$. The blue curve shows the profile closest to the Northern Escanaba (NESCA) lava flow. The profiles for the 8 surrounding profiles (on a grid with a resolution of 0.5 degrees) are also shown as gray curves, illustrating the near-uniformity of the profiles of $N$ over the entire region. The plots show that the value $N \approx 10^{-3}$ s$^{-1}$ characterizes the abyssal ocean, which we utilized in our inversion for the rate of heat transfer $\Phi$ in Eq. (4). (**B**) shows a plan-view of the sampled coordinates, with the position of the NESCA lava flow indicated by the yellow star.

## Data availability
The data analyzed in this study is published in[15].

## Code availability
Custom computer codes used to generate the results reported in this paper are available from the corresponding author upon reasonable request.

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

## Author contributions

S.S.P. developed the mathematical model and conducted the inversion. D.J.F. developed the geological interpretation and analysis. Both authors planned the study and wrote the paper.

## Competing interests

The authors declare no competing interests.
