## [Peer Review File · Nature Communications]

REVIEWER COMMENTS

Reviewer #1 (Remarks to the Author):

Comments on "Rapid heat discharge during deep-sea eruptions generates megaplumes and disperses tephra (Samuel S. Pegler and David J. Ferguson)"

Reviewer: Mathieu Colombier

Main comment: This study aims to shed light on the processes causing megaplume formation observed in deep oceans and commonly correlated with mid-oceanic ridge volcanic activity. The authors use an inverted methodology to retrieve the amount and sources of heat responsible for the formation of such megaplumes and causing the dispersal of tephra at distances up to 1km. They show that effusive/explosive volcanic activity alone cannot explain it and that an additional role of intracrustal hydrothermal fluids is a necessary condition for the formation of megaplumes.

I found the manuscript very interesting, well-written and innovative. The authors considered most of the possible aspects that could influence their model and discuss it in a convincing manner. I would suggest acceptance of the manuscript after some minor revisions.

I have mostly 2 comments that I would like to be addressed in the next version of the manuscript.

Main comment 1: I find that the authors give too much importance to the nature of the deep sea submarine eruptions (effusive vs. explosive). Although it has been proposed that explosive eruptions (causing pyroclastic deposits) can occur in these deep submarine settings (e.g., the study of Clague et al that serves here as a basis for the authors model), this topic remains still a bit controversial. For example, Schipper et al. 2013 (doi:10.1093/gji/ggs099) showed that the tephra commonly observed at the sea floor in these deep settings (i.e., Limu o Pelé particles) can be reproduced experimentally by non-explosive thermal granulation. Although we have direct evidence of explosive activity for instance for Rota volcano or West Mata (with videos of the activity), these correspond to intermediate depth compared to the deep setting discussed in this study. Further in the method section, the authors also acknowledge in line 98 that thermal granulation is a possible process and that it does not influence their models. Although I personally believe that deep submarine explosive activity is possible, I would suggest that the authors now present the different possible causes for the formation of fine particles in these deep sea environments (explosive magmatic activity vs. thermal granulation during principally effusive activity) instead of saying that these particles are surely formed by explosive activity. Some modifications of the text in this regard could be done in lines 57-74.

Main comment 2: on the section "Sources of megaplume energy"

What about an additional effect of lava degassing as a source of heat? Volcanic gas can escape efficiently through low viscosity basaltic magmas. Therefore, a high amount of hot volcanic gases could be released directly above the lava flow (and mostly at the source) and feed the megaplume. Maybe I missed it but I think you could discuss this process also. So then the sources of heat would be (i) the cooling lava flow + potential pyroclasts, (ii) the intracrustal fluids heated by the intrusive and extrusive lava and (iii) the volcanic gases percolating through the lava

Main comment 3: If it is true that there was explosive activity, I imagine that the violent expulsion of tephra and gas mixture at high speed would have provided additional energy (in addition to buoyancy) for the rise of the megaplume and subsequent tephra dispersal. Did you take this into account?

L38

Exsolved volatiles could also come from the effusive lava, not only from the pyroclasts... In fact, volatile exsolution and gas escape from the magma is more important during effusive activity. That could be an additional source of heat and fluids. See the main comment 2 related to this aspect.

L49

"first inversion of a submarine tephra deposit". I would rephrase this differently. We do not "invert" a deposit.

L56

See my main comment related to this topic.

L63

These studies deal with depths of 550–560 m for NW Rota-1 volcano and 1,200 m depth at West Mata so we are still far from the deep sea setting discussed here. In addition, I would say these explosive eruptions were not so energetic and intermediate between explosive and effusive in the case of West Mata. As stated in the main comment, the authors should to my opinion reformulate this section in a way that does not rule out explosive eruptions in deep sea, but with some other alternatives for the formation of fine particles such as thermal granulation.

L78

Here and elsewhere in the manuscript, I don't like too much the term "inversion of tephra", I find it very vague. Is that an accepted terminology in the literature?

Also please rephrase to "a model of neutrally buoyant umbrella..."

L94

"1000 μm " Replace by 1mm

L97

In relation to previous comments, when I look at the tephra described in the Clague et al studies, many of the particles could be produced by effusive activity plus thermal granulation. I recognize that this is harder to explain the Pele's hair with such a scenario. Maybe we have both processes taking place (explosive magma fragmentation + thermal granulation)

L98

Good that the authors mention this process. I think this possibility could be presented earlier in the manuscript.

L141

Here and elsewhere: the term intrusion is a bit confusing as we are also dealing with magmatic intrusions. Is there a more appropriate terminology?

L141

km-sale: correct to km-scale

L186

"The stem of the plume is modelled as a turbulent, vertically convecting column of hot water"

What would be the influence of tephra, volcanic gas and hydrothermal fluids on this simplified model?

L249

change to eruption timescale

L262

Can the authors tell a bit more about this cooling model and its assumptions in the main text?

L263

What if a large fraction of this crust is then rapidly removed via thermal granulation for instance?

L265

Where does this 30% come from?

L269-272

This is still in the same order of magnitude, the difference only caused by the 30 % assumption.

L281-285

This sentence is hard to follow. Do you mean the fluids must come from the erupted magma rather than the crust? Please clarify.

L292

I would not say "likely dominant"

L299

"so far"?

L429

change to "it remains only to estimate the"

Kind regards,
Mathieu Colombier

Reviewer #2 (Remarks to the Author):

Review of "Rapid heat discharge during deep-sea eruptions generates megaplumes and disperses tephra" by S. Pegler and D. Ferguson.

Event plumes have been puzzling since their discovery was reported in Nature in 1987. They represent massively large inputs of hydrothermal fluid, and perhaps biota, but the importance of that supply depends on their unknown frequency. The mechanism of EP creation has defied easy explanation, largely because the duration of the formation process remains uncertain, with published estimates (mostly guesses) ranging from hours to weeks.

This paper provides a well-written and novel explanation of EP formation based on the dispersal of lava tephra on the seafloor surrounding an eruption. The authors offer a convincing (with some exceptions) model of how buoyancy-driven flow in the plume "umbrella" produces a tephra distribution that constrains the energy discharge rates and duration of an EP event. They conclude that an EP

presumably associated with the NESCA lava flow on the Gorda Ridge formed in ~ 12 hr, consistent with an earlier model based on a different EP by Lavelle (1995) using an entirely different methodology. This is (to my knowledge) the first paper with a quantitative model finding a rapid EP formation (< 1 day) from a combination of lava cooling (minor) and release of hot fluids from the crust (major). Other models have been largely qualitative or incorrect because cooling of extruded pillow lavas or an intruded dike cannot power a rapid EP formation. The authors also note that observations of tephra distributions may lead to the identification of "fossil" EP sites.

The finding that most of the heat must come from intra-crustal fluids also helps explain the unique EP chemistry. $^3\text{He}/\text{heat}$ ratios of EPs are extremely and uniformly low, somewhat similar to "mature" hydrothermal discharge. These ratios, as well as consistent Mn/heat ratios, suggest the fluids in EPs are predominantly formed by the release of pre-existing hydrothermal fluids, not by rapid heating of non-hydrothermal crustal fluids by an intruding dike (see also Lupton et al., 1999, EPSL). The paper also suggests an alternative fluid source by "thermomechanical effects of dyke emplacement" after [9], but the Lowell and Germanovich model requires a lengthy heat transfer time on the order of ~ 10 days.

Another EP mystery is the common appearance of multiple plumes from a single dike intrusion. Is the model consistent with, especially, the apparently rapid creation of several small EPs, of varying energy (rise height) during the 2008 Lau basin eruption?

The paper is well written and organized, the Supplementary Information is extensive and (mostly) thorough. I think the paper will be of interest to a wide audience interested in seafloor processes, as EP creation is a fundamental, but poorly understood, feature of the exchange of heat, material, and biota between the earth's solid and liquid phases. Please note that I am unable to evaluate all of the mathematical reasoning in the paper.

I do have two issues with the paper: stem geometry and time duration of formation.

Stem geometry: A critical issue is resolving the conflict between the idealized radial plume stem assumed in the paper and prior observational and model evidence for planar EP plume stems. The paper uses a "top-hat" discharge model that explicitly assumes a stem with a radial geometry, although this geometry is not consistent throughout the paper. For example, Fig 2 shows the modeled NESCA source location as a point nearly coincident with the location of the "eruptive vent" of the lava flow, as for the "top hat" model, and consistent with an "axisymmetric" source implied by the observed seafloor tephra dispersion pattern. This model is normally used for discharge from a conventional vent chimney a few cm in diameter. Conversely, Fig 3 visually describes the stem of an EP as a plane aligned along ~ 2 km of an eruptive fissure (unless that is meant to show a radial plume of 2 km diameter, which is unrealistic). The planar model is in agreement with the EP model of Lavelle (1995), based on one of the CoAxial EPs. Lavelle found it required a generation time of ~ 2 hr for a fissure length of 1200 m. A generation time of 1 hr thus implies a fissure length of ~ 2400 m, roughly the length of the emplaced CoAxial lava flow. Baker et al. (1989) used a similar model (also by Lavelle) to argue for the requirement of a linear source, as opposed to the unreasonable assumption of a circular vent of many meters. The planar plume model is also consistent with many observations of seafloor (and terrestrial) eruptions that extend for kms along a fissure. The paper needs to explicitly deal with these apparent contradictions in stem geometry. I wonder if there is any possibility that the higher scatter in tephra mass dispersion along the fissure strike (Fig. 3) could be related to a planar, not axisymmetric, stem?

Time duration: The value of 12 hr for an EP discharge (which I consider reasonable) is inadequately supported. In lines 247-250 the paper uses a lava effusion rate of $10 \text{ m}^3/\text{s}$ to derive an eruptive time scale of ~ 12 hr. [23] says that NESCA is a mixture of sheet flows and pillows; pillows probably later in

the eruption. [48] gives effusion rates of 1-100 m³/s for lobate sheet flows, faster for other kinds, slower for pillows. If 1 m³/s, then the time scale is 120 hrs, but if 100 m³/s then 1.2 hrs (correct?). Both extremes (especially 120 hrs) seem unsuited to the model (and observations), and so 10 m³/s seems chosen simply to benefit the model. This is not necessarily improper, but the choice should be justified. Moreover, Table 1 says the eruption duration is inferred from both the effusion rate and the lava volume, but the lava volume doesn't seem to have been considered in the text. Based on instruments caught within the 1998 lava flows on Axial volcano, Fox et al. (2001) calculated a maximum lava effusion rate of 2.7 m³/s, declining to <1 m³/s, for the thin sheet flows of this eruption (although no true EP was observed, perhaps due to the absence of any observations until 18 d post-eruption). The 1996 Gorda Ridge eruption (not NESCA), on the other hand, was primarily pillow flows, suggesting an effusion rate of 1-10 m³/s (Chadwick et al. 1998), more in line with your choice of 10 m³/s.

Edward T. Baker

Reviewer #3 (Remarks to the Author):

I think that this is a well-written paper worthy of publication. The topic, mechanics of the dispersal of submarine tephra by megaplumes, is timely and of widespread interest. The authors do an excellent job using deposits to evaluate quantitative models and estimate eruption characteristics (e.g., heat flux) that are hard to directly measure. Quantitative constraints of MOR eruption dynamics are especially valuable given that MOR eruptions have not been visually witnessed. I think that this ms has the potential to become a well-referenced classic. Their quantitative methods seem sound, albeit if I don't have the expertise to evaluate all the details (e.g., the tidal model). I have comments and suggestions below. I do disagree with the authors' conclusion that intracrustal fluid release is needed to generate tephra-transporting megaplumes. Instead, I think their data suggests that lava flow heating is the energy source for the plume studied here.

Review of “Rapid heat discharge during deep-sea eruptions generates megaplumes and disperses tephra” by Peglar and Ferguson

Summary

I think that this is a well-written paper worthy of publication. The topic, mechanics of the dispersal of submarine tephra by megaplumes, is timely and of widespread interest. The authors do an excellent job using deposits to evaluate quantitative models and estimate eruption characteristics (e.g., heat flux) that are hard to directly measure. Quantitative constraints of MOR eruption dynamics are especially valuable given that MOR eruptions have not been visually witnessed. I think that this ms has the potential to become a well-referenced classic. Their quantitative methods seem sound, albeit if I don't have the expertise to evaluate all the details (e.g., the tidal model). I have comments and suggestions below. I do disagree with the authors' conclusion that intracrustal fluid release is needed to generate tephra-transporting megaplumes. Instead, I think their data suggests that lava flow heating is the energy source for the plume studied here.

Major comments:

More specific information on the observational constraints of megaplumes would be helpful (e.g., lines 31-35, lines 311-312). For example, How many megaplumes have been observed and with what instrumentation? Have volcanoclastic particles been observed in active megaplumes? What other evidence suggests that they correlate with volcanic eruptions? What is known about how long megaplume events last?

The assessment of the source of the thermal energy for the megaplume could be expanded upon. The estimates given on lines 261-280 show that lava flows provide approximately the same amount of energy as is estimated for the megaplume, leading me to think lava flows are a very plausible energy source. I see that you say that only 30% of the lava thermal energy goes into the plume, however I am unconvinced by this brief statement. The lava will eventually cool to ambient temp and release its thermal energy. Because there is no mention of time in these calculations, I don't see why having a crust slow the rate of heat transfer is relevant to the calculation? One would need to estimate Q from the flow rather than the total amount of energy released. Even if 30% efficiency is right, the lava flow and plume energy estimates are still within a factor of 3, which seems fairly close to me given that there are other uncertainties that aren't fully explored here.

I would also appreciate more information about the nature of these intracrustal fluids. What are they? What do we know about them? Why are they suddenly released during a tephra producing submarine eruption? To me it seems arbitrary to invoke the release of additional fluids of unknown origin without a mechanism.

I think that the ms could be clearer about the observations and model assumptions regarding particle size. That is, did the Clague et al., study observe trends in particle size? Did you use a

mean particle size from their observations? From the main text, I think that you only model one particle size (that is, fall speed), however it is unclear. There is a section on polydisperse deposition in the supplement, however it was unclear how this connected to the assumptions and treatment in your model.

The characteristic length scale, L , of the deposit is central to your model. I didn't understand how L physically related to the actual Clague et al., deposits until reading the supplement (that is, fitting equation 2 to the tephra profile). I suggest you are more clear about the physical meaning of L in the main text. Some readers might think L is where you suddenly stop finding tephra (which is not the case).

A short statement related to your assumptions or treatment of the steadiness of the eruption could be helpful. Your model assumes that eruptions are steady, but in reality eruptions may not be. How does unsteadiness affect the ability to interpret deposits? How would one recognize a steady vs. unsteady eruption from the Clague et al. deposits?

Line comments:

Lines 77 – 86: I think that authors are correct in using a model of a spreading gravity current. They nicely explain that this is different than what is typically used for subaerial plumes. I think a brief explanation of why they choose a different type of model (compared to subaerial plumes) would be helpful for the readers on these lines. That is, in what ways does the submarine setting differ from the atmospheric setting to motivate a different kind of model? What basic assumptions go into their choice of model? (I now see that they have this farther down in the ms).

Lines 55-56: I disagree that the model shows “direct evidence” of a connection. Rather, the model results show that the energy required for pyroclast dispersal are the same/similar to the energy required for megaplumes. To me, direct evidence of a connection would be the observation of volcanoclastic particles within megaplumes. I am curious if this has been reported previously in the literature?

Line 57-58: I find the term “highly unlikely” somewhat dissatisfying. Instead, can you make quantitative comparisons of the energy discharge. 1TW (need a heat loss model for lavas to see if one can lose heat quickly enough)

Line 76: Would it be possible to specify the composition of the “volume flux.”

Lines 95-96: Maybe a statement on how Clague et al. knew that the volcanoclastic particles came from a singular and same eruption?

Line 108-110: I'm not sure I understand.

Line 115: Thinning of mass, but what about particle size sorting?

Line 140: Can you specify what you mean by “which”?

Line 148-149: what about geostrophic flow?

Figure 3: Add seawater entrainment.

Label the pink part of the plume as a mixture of particles and warm fluid

Include an arrow in the central part of the plume to indicate upward flow.

Perhaps put an arrow to indicate downward particle settling below the plume

I don't think I understand tidal sway

Put figure 3 before figure 2.

160: Perhaps say “elongated deposit” rather than “near-linear plume”

Line 197: Maybe reference how particle size comes into the settling speed.

Also you state your assumptions about how the particles are coupled to the flow.

Line 198: Can you specific about what is in this volume flux (mixture of particles and fluid?)

Line 202: Could this equation be tested by comparing dispersal distances for different size particles within a single eruption? What if Q is not constant in time?

Line 208: How is the dispersal scale, L , compared to measurements on the ground?

Line 223: Does this fitting mean that you need more than one observational profile to identify a source vent. Or, could you ID a vent with just one profile? Maybe you could comment on how more than one profile is needed to assess whether currents play a role in dispersal (what errors would be associating your model with a profile that was advected by currents).

Line 248-249: It is not clear to me how you infer an eruptive timescale from an eruptive flux (that is, how is volume estimate for this calculation?).

Line 280: I think that this language is too strong. Your results “suggest” fluid release, but I don't think prove it.

Line 282: A better definition of this crustal fluid could be helpful. That is, how do you define a hydrothermal reservoir vs. heated pore fluid?

Line 287: Can you tell me what these microbes are and why they are associated with hydrothermal fluid?

Line 291: Here you mention the speed of heat transfer. However, nothing previously in your analysis suggested anything about speed.

Line 355-356: I disagree with this interpretation. You show that data fits a model.

Line 429: Typo, It remains "to"?

Line 433: Replace "bed source" with "base."

Equation 5: I did not verify all the details in these equations. However, I like your approach. One comment is that this treatment doesn't include the energy required to loft particles upward. That is, it is a one phase treatment of a plume. A sentence on the assumptions going into this model could be helpful as well as why you can neglect the multiphase nature of the plume in the stem.

I don't have expertise to evaluate the accuracy of the tidal model.

I had a hard time following the reaching the umbrella section. I think the coupling of the fluid and particles is important and am glad this section exists. It might help if you describe better the physical meaning of the dimensionless group in this section. How does it relate to other parameters like the Rouse number that are commonly used for fluid particle settling? Why do you use this dimensionless parameter?

Reviewer #4 (Remarks to the Author):

Review for NCOMMS-20-23222

“Rapid heat discharge during deep-sea eruptions generates megaplumes and disperses tephra”
by Pegler and Ferguson.

The paper details the creation of a relatively simple mathematical model to account for the submarine transport and deposition of tephra occurring from the buoyant spread of a tephra from an umbrella-shaped distribution and use the model to quantify energy release that occurs at the seafloor. A rival hypothesis stating that observed distribution can be the effect of tidal currents is also modelled and shown to lead to inappropriate solutions.

The paper is well-written and is easy to follow. The methodology seems simple on the surface but the model is based on physically and mathematically appropriate choices, worked from a foundational point to lead to a Gaussian solution, rather than fitting a Gaussian solution a priori. I feel that this justification of the methodology is one of the strong points of the paper.

My main issues concern the: (i) lack of any quantified error analysis on the final fit (and partially as a consequence stemming from this, the lack of errors in the estimated values for other parameters), and (ii) the seemingly arbitrary rejection of one of the four profiles used. As this is the first time the proposed model is used I feel that the fitting errors should be presented in detail, making the inclusion of bad fits especially relevant (at least in the error analysis). On a more unrelated note I think that the manuscript can be reorganised to better utilise the nature publication structure style (i.e. methods section at the end of the paper). I would argue against including the mathematical foundations of the paper as supplemental material as in my opinion it is necessary information to evaluate the study. These and several smaller points can be found in the attached PDF document.

Overall I believe that the paper fits the scope of the journal, has an important and novel point, potential impact, and merits publication. I wish that all papers that are sent to me to review were of this standard. However, I still believe that some aspects should be critically reviewed before publication. I hope that the authors find the points raised constructive.

Kind regards and good luck with the revisions.

Responses to Reviewer 1

We thank the referee for their constructive review and many helpful and positive comments. We are pleased that the referee considers the work innovative, interesting, well written, and that acceptance is recommended after minor revisions. We agree with the vast majority of comments which have led to numerous improvements in revision. Please find our responses to the comments below and descriptions of the revisions we have made. Please find revised text shown in blue in the new manuscript.

Main comment: This study aims to shed light on the processes causing megaplume formation observed in deep oceans and commonly correlated with mid-oceanic ridge volcanic activity. The authors use an inverted methodology to retrieve the amount and sources of heat responsible for the formation of such megaplumes and causing the dispersal of tephra at distances up to 1km. They show that effusive/explosive volcanic activity alone cannot explain it and that an additional role of intracrustal hydrothermal fluids is a necessary condition for the formation of megaplumes. I found the manuscript very interesting, well-written and innovative. The authors considered most of the possible aspects that could influence their model and discuss it in a convincing manner. I would suggest acceptance of the manuscript after some minor revisions.

Thank you for these comments, and for suggesting acceptance following minor revisions.

I have mostly 2 comments that I would like to be addressed in the next version of the manuscript.

Main comment 1: I find that the authors give too much importance to the nature of the deep sea submarine eruptions (effusive vs. explosive). Although it has been proposed that explosive eruptions (causing pyroclastic deposits) can occur in these deep submarine settings (e.g., the study of Clague et al that serves here as a basis for the authors model), this topic remains still a bit controversial. For example, Schipper et al. 2013 (doi:10.1093/gji/ggs099) showed that the tephra commonly observed at the sea floor in these deep settings (i.e., Limu o Pelé particles) can be reproduced experimentally by non-explosive thermal granulation. Although we have direct evidence of explosive activity for instance for Rota volcano or West Mata (with videos of the activity), these correspond to intermediate depth compared to the deep setting discussed in this study. Further in the method section, the authors also acknowledge in line 98 that thermal granulation is a possible process and that it does not influence their models. Although I personally believe that deep submarine explosive activity is possible, I would suggest that the authors now present the different possible causes for the formation of fine particles in these deep sea environments (explosive magmatic activity vs. thermal granulation during principally effusive activity) instead of saying that these particles are surely formed by explosive activity. Some modifications of the text in this regard could be done in lines 57-74.

We now represent the alternative proposed mechanisms for clast generation. While the specific process is indeed not central to our paper (we only require ash to be created for use as a flow

tracer), we agree with the referee that it is important to convey the debate in regard to clast generation:

Revised introductory text:

“Although some debate exists on whether these tephra are generated primarily via magmatic fragmentation of fluid magma [16] versus other brecciation processes such as thermal granulation [32] or hydrovolcanic fragmentation [31], many authors have interpreted these deposits as evidence for explosive volcanism occurring in the deep ocean [16, 19]...”

Main comment 2: on the section “Sources of megaplume energy”. What about an additional effect of lava degassing as a source of heat? Volcanic gas can escape efficiently through low viscosity basaltic magmas. Therefore, a high amount of hot volcanic gases could be released directly above the lava flow (and mostly at the source) and feed the megaplume. Maybe I missed it but I think you could discuss this process also. So then the sources of heat would be (i) the cooling lava flow + potential pyroclasts, (ii) the intracrustal fluids heated by the intrusive and extrusive lava and (iii) the volcanic gases percolating through the lava

We have substantially re-written this section and now include - in addition to a more quantitative approach to assessing lava heating - a simple evaluation of the possible role of exsolved volatiles. While it does seem to be the case that a separate CO₂ rich fluid phase can more readily supply the rate of heat transfer necessary, the mass flux of CO₂ required is extremely high. We now describe it as a possible explanation, however further work on eruptive processes are required to fully evaluate its feasibility.

Also added to discussion at bottom of page 15 in summary:

“It may also be the case that the interaction of ascending magmas with CO₂ rich foams is more prevalent during MOR eruptions than previously thought and that the initial phase of seafloor eruptions are often characterized by high rates of energy transfer via exsolved volatiles.”

Main comment 3: If it is true that there was explosive activity, I imagine that the violent expulsion of tephra and gas mixture at high speed would have provided additional energy (in addition to buoyancy) for the rise of the megaplume and subsequent tephra dispersal. Did you take this into account?

Initial momentum generated by explosive activity will be rapidly dissipated by the inertia of the ambient seawater. As a case in point, a bullet fired at 100s m/s into water will reach just 1 m before the inertia of the water dissipates the speed entirely. For the much smaller particles we are concerned with, and with velocities generated at the base of the volcanic plume from explosive activity orders of magnitude smaller, the contribution of initial kinetic energy to particle migration will be considerably less than one metre, and can thus be safely neglected in considering several km of dispersal.

Line Comments

L38 Exsolved volatiles could also come from the effusive lava, not only from the pyroclasts... In fact, volatile exsolution and gas escape from the magma is more important during effusive activity. That could be an additional source of heat and fluids. See the main comment 2 related to this aspect.

We refer to our response to main comment 2 and the revised text in the relevant section.

L49 “first inversion of a submarine tephra deposit”. I would rephrase this differently. We do not “invert” a deposit.

Revised to “inversion of dispersal data from a submarine tephra deposit”

L56 See my main comment related to this topic.

We refer to our response to main comment 1 above.

L63 These studies deal with depths of 550–560 m for NW Rota-1 volcano and 1,200 m depth at West Mata so we are still far from the deep sea setting discussed here. In addition, I would say these explosive eruptions were not so energetic and intermediate between explosive and effusive in the case of West Mata. As stated in the main comment, the authors should to my opinion reformulate this section in a way that does not rule out explosive eruptions in deep sea, but with some other alternatives for the formation of fine particles such as thermal granulation.

Agreed (see our response to main comment 1 and revisions described there).

L78 Here and elsewhere in the manuscript, I don't like too much the term “inversion of tephra”, I find it very vague. Is that an accepted terminology in the literature?

Revised throughout to be specific about what we are inverting (tephra data).

Also please rephrase to “a model of neutrally buoyant umbrella...”

Revised as suggested.

L94 “1000 μm ” Replace by 1mm

Revised as suggested.

L97 In relation to previous comments, when I look at the tephra described in the Clague et al studies, many of the particles could be produced by effusive activity plus thermal granulation. I recognize that this is harder to explain the Pele's hair with such a scenario. Maybe we have both processes taking place (explosive magma fragmentation + thermal granulation)

We have removed “during explosive discharge” to leave open the possibility that the clasts were generated by other mechanisms such as thermal granulation. The possibility of both of these processes for clast generation is now mentioned earlier in the paper (addressed under our response to main comment above).

L98 Good that the authors mention this process. I think this possibility could be presented earlier in the manuscript.

Please see our responses clarifying the general possibility of other processes for tephra production above.

L141 Here and elsewhere: the term intrusion is a bit confusing as we are also dealing with magmatic intrusions. Is there a more appropriate terminology?

Agreed - generally **removed** the term intrusion in referring to the umbrella regime (with a single exception as a parenthetical note to provide a helpful highlighting of this terminology for the fluid-mechanical audience, where it is standardised for this regime of flow):

“as a turbulent, primarily horizontally flowing neutrally buoyant gravity current (or intrusion), forming the umbrella of the plume”

L141 km-sale: correct to km-scale

Corrected.

L186 “The stem of the plume is modelled as a turbulent, vertically convecting column of hot water” What would be the influence of tephra, volcanic gas and hydrothermal fluids on this simplified model?

Included a detailed discussion of the concentration of tephra in the plume system in section 1 of the Supplementary Methods. Hydrothermal is incorporated.

L249 change to eruption timescale

Revised as suggested here and elsewhere

L262 Can the authors tell a bit more about this cooling model and its assumptions in the main text?

We now utilise the results of a theoretical model of heat conduction to predict the heat flux from cooling magma (Dechamps et al. 2010).

L263 What if a large fraction of this crust is then rapidly removed via thermal granulation for instance?

We understand the suggestion but note that the observed volumes of fragmental material versus intact lava at this and other submarine eruptions suggests that such substantial (and continued) removal of the quenched crusts does not occur (in order to account for the total energy content required, fragmentation of all (or almost all) of the erupted lava would be required).

L265 Where does this 30% come from?

We now present a detailed analysis comparing available fluxes (as opposed to total energies) which does not depend on this assumption. For this, the subsection titled “A direct volcanic origin for megaplumes?” has been expanded and reworked throughout. The appeal to fluxes directly is done by comparing the energy flux we infer (bypassing also the uncertainty of the eruption duration) with an upper bound on energy flux per unit area from a time-dependent

thermodynamic model of cooling submarine lava. The result indicates that heating by cooling lava is insufficient to provide the required energy flux.

L269-272 This is still in the same order of magnitude, the difference only caused by the 30 % assumption.

See comment immediately above.

L281-285 This sentence is hard to follow. Do you mean the fluids must come from the erupted magma rather than the crust? Please clarify.

In revising this section we have reassessed the likelihood of *in situ* heating of pore fluid by the dyke and no longer present it as a likely mechanism.

L292 I would not say “likely dominant”

We have now revised this subsection and this phrase no longer appears.

L299 “so far”?

Changed as suggested.

L429 change to “it remains only to estimate the”

Changed as suggested.

Responses to Reviewer 2

Event plumes have been puzzling since their discovery was reported in Nature in 1987. They represent massively large inputs of hydrothermal fluid, and perhaps biota, but the importance of that supply depends on their unknown frequency. The mechanism of EP creation has defied easy explanation, largely because the duration of the formation process remains uncertain, with published estimates (mostly guesses) ranging from hours to weeks.

This paper provides a well-written and novel explanation of EP formation based on the dispersal of lava tephra on the seafloor surrounding an eruption. The authors offer a convincing (with some exceptions) model of how buoyancy-driven flow in the plume “umbrella” produces a tephra distribution that constrains the energy discharge rates and duration of an EP event. They conclude that an EP presumably associated with the NESCA lava flow on the Gorda Ridge formed in ~12 hr, consistent with an earlier model based on a different EP by Lavelle (1995) using an entirely different methodology. This is (to my knowledge) the first paper with a quantitative model finding a rapid EP formation (<1 day) from a combination of lava cooling (minor) and release of hot fluids from the crust (major). Other models have been largely qualitative or incorrect because cooling of extruded pillow lavas or an intruded dike cannot power a rapid EP formation. The authors also note that observations of tephra distributions may lead to the identification of “fossil” EP sites.

The finding that most of the heat must come from intra-crustal fluids also helps explain the unique EP chemistry. $^3\text{He}/\text{heat}$ ratios of EPs are extremely and uniformly low, somewhat similar to “mature” hydrothermal discharge. These ratios, as well as consistent Mn/heat ratios, suggest the fluids in EPs are predominantly formed by the release of pre-existing hydrothermal fluids, not by rapid heating of non-hydrothermal crustal fluids by an intruding dike (see also Lupton et al., 1999, EPSL). The paper also suggests an alternative fluid source by “thermomechanical effects of dyke emplacement” after [9], but the Lowell and Germanovich model requires a lengthy heat transfer time on the order of ~10 days.

Another EP mystery is the common appearance of multiple plumes from a single dike intrusion. Is the model consistent with, especially, the apparently rapid creation of several small EPs, of varying energy (rise height) during the 2008 Lau basin eruption?

The paper is well written and organized, the Supplementary Information is extensive and (mostly) thorough. I think the paper will be of interest to a wide audience interested in seafloor processes, as EP creation is a fundamental, but poorly understood, feature of the exchange of heat, material, and biota between the earth’s solid and liquid phases. Please note that I am unable to evaluate all of the mathematical reasoning in the paper.

I do have two issues with the paper: stem geometry and time duration of formation.

We thank the referee for the detailed and positive review of our submission.

We are pleased to read that the referee considers the paper to be well-written and organised, novel, and of wide interest. We fully understand the two points, and have made revisions in light of them both that we believe have led to significant improvement. Please find our responses below and descriptions of our revisions, and please find the revised text shown in blue in the revised manuscript.

Stem geometry: A critical issue is resolving the conflict between the idealized radial plume stem assumed in the paper and prior observational and model evidence for planar EP plume stems. The paper uses a “top-hat” discharge model that explicitly assumes a stem with a radial geometry, although this geometry is not consistent throughout the paper. For example, Fig 2 shows the modeled NESCA source location as a point nearly coincident with the location of the “eruptive vent” of the lava flow, as for the “top hat” model, and consistent with an “axisymmetric” source implied by the observed seafloor tephra dispersion pattern. This model is normally used for discharge from a conventional vent chimney a few cm in diameter. Conversely, Fig 3 visually describes the stem of an EP as a plane aligned along ~2 km of an eruptive fissure (unless that is meant to show a radial plume of 2 km diameter, which is unrealistic). The planar model is in agreement with the EP model of Lavelle (1995), based on one of the CoAxial EPs. Lavelle found it required a generation time of ~2 hr for a fissure length of 1200 m. A generation time of 1 hr thus implies a fissure length of ~2400 m, roughly the length of the emplaced CoAxial lava flow. Baker et al. (1989) used a similar model (also by Lavelle) to argue for the requirement of a linear source, as opposed to the unreasonable assumption of a circular vent of many meters. The planar plume model is also consistent with many observations of seafloor (and terrestrial) eruptions that extend for kms along a fissure. The paper needs to explicitly deal with these apparent contradictions in stem geometry. I wonder if there is any possibility that the higher scatter in tephra mass dispersion along the fissure strike (Fig. 3) could be related to a planar, not axisymmetric, stem?

We agree and have now expanded our theoretical analysis to include the predictions of a planar stem model. The result is a more generalized inversion approach that allows for the possibilities of both short and long fissure lengths, with predictions that remain consistent with our previously inferred source buoyancy flux over a range of possible fissure lengths (up to a few km in fissure length). Though a point source model, being representative for $l < 1$ km (this scale is discussed below), is plausible in view of the bathymetry showing a central ring fault, it is helpful to consider how the results change in view of a longer fissure length. Our revisions constitute new material [see new text spanning Eqn.s (9) to (13) in section 1 of the Supplementary Methods], and new text in the main paper, as detailed below.

In summary, we now allow theoretically for all possible situations (spanning all ranges of fissure lengths, source heat fluxes and ocean stratification) encompassing the following: (i) a point-source (axisymmetric) stem model representing a short fissure (how short will be detailed below; the geometrical details of the source of a plume become unimportant if smaller than an

intrinsic scale l_* that we define), (ii) a planar model, equivalent to an idealised 2D line-source model for sufficiently long fissures, or (iii) some intermediate situation between the two if fissure length $l \sim l_*$. In the new Fig. 5, we show our inverted source buoyancy flux as a function of fissure length, with the characteristic switchover length scale being at $l_* \sim 1200$ m, in accordance with the prediction based on characteristic inferred values (at this value, the predictions of both the axisymmetric model and the planar model are equivalent). A fully resolved 3D DNS simulation would predict a similar plot, with the only change being a slightly smoothed curve between the asymptotes (and agree exactly in the respective endmember limits). As shown by Fig 5, the model predictions for the inferred buoyancy source flux decrease as fissure length is increased, which occurs because a planar source is more efficient at entraining ambient seawater (for the same buoyancy flux) due to the larger surface area along the sides of the stem. The upper bound resulting from our maximum inferred umbrella flux, referred to as $Q_{umb}^{(max)}$, results in a smaller upper bound on our inferred F_0 that lies within our existing range of inferred values from the point-source model. The minimum $Q_{umb}^{(min)}$ results in a lower F_0 than the point-source model (to a value of $130 \text{ m}^4/\text{s}^3$ for an example fissure length of 4 km).

Revision: The above theory and points of discussion are now provided in section 1 of the Supplementary Methods.

Added: A new figure (Figure 5, copied here, right) to the Supplementary Methods showing the predicted source buoyancy flux as a function of fissure length predicted by the unified model and the characteristic partitioning between the theories along the line where they are coincident.

Also **Revised** the schematic Fig. 3 to show more realistic proportions.

Time duration: The value of 12 hr for an EP discharge (which I consider reasonable) is inadequately supported. In lines 247-250 the paper uses a lava effusion rate of 10 m³/s to derive an eruptive time scale of ~12 hr. [23] says that NESCA is a mixture of sheet flows and pillows; pillows probably later in the eruption. [48] gives effusion rates of 1-100 m³/s for lobate sheet flows, faster for other kinds, slower for pillows. If 1 m³/s, then the time scale is 120 hrs, but if 100 m³/s then 1.2 hrs (correct?). Both extremes (especially 120 hrs) seem unsuited to the model (and observations), and so 10 m³/s seems chosen simply to benefit the model. This is not necessarily improper, but the choice should be justified. Moreover, Table 1 says the eruption duration is inferred from both the effusion rate and the lava volume, but the lava volume doesn't seem to have been considered in the text. Based on instruments caught within the 1998 lava

flows on Axial volcano, Fox et al. (2001) calculated a maximum lava effusion rate of 2.7 m³/s, declining to <1 m³/s, for the thin sheet flows of this eruption (although no true EP was observed, perhaps due to the absence of any observations until 18 d post-eruption). The 1996 Gorda Ridge eruption (not NESCA), on the other hand, was primarily pillow flows, suggesting an effusion rate of 1-10 m³/s (Chadwick et al. 1998), more in line with your choice of 10 m³/s.

We have refined our estimate of the likely duration of the eruptive period constrained by our modelling (paragraph from top of page 12) and now allow for a range of eruptive periods between 10-20 hours. This is based on estimating the portion of the erupted lava that was probably associated with pyroclast formation (and therefore the period of hydrothermal discharge that is actually constrained by our dispersal model) rather than a crude estimate of the total duration. Observations suggest that deep-sea tephra is produced during high-effusion rate sheet-flow forming activity. Based on current observations, it is thought that sheet-flows comprise around a third of the erupted volume at NESCA (high-res bathymetric data collected by D Clague is currently being analyzed and will refine this). Using a range of typical volumetric discharge rates for ropy sheet-flows of 200-500 m³ s⁻¹ gives an eruptive period of 10-20 hours. Although we believe this estimate is more robust, it is important to note that our conclusion that the tephra was transported in a megaplume does not strongly depend on the duration chosen, as any reasonable value results in a plume with an energy content akin to a megaplume (Fig. 4).

Responses to Reviewer 3

We thank the referee for their constructive and detailed review and many positive comments supporting publication. We were pleased to read that the referee considers the work to be worthy of publication, well-written, timely, and has the potential to be a well-referenced classic.

We agree with the vast majority of the referee's suggestions and have made corresponding revisions. These have led to numerous improvements in revision. Please find our responses to each of the comments below and descriptions of the revisions we have made, and please find revised text shown in blue in the new manuscript.

I think that this is a well-written paper worthy of publication. The topic, mechanics of the dispersal of submarine tephra by megaplumes, is timely and of widespread interest. The authors do an excellent job using deposits to evaluate quantitative models and estimate eruption characteristics (e.g., heat flux) that are hard to directly measure. Quantitative constraints of MOR eruption dynamics are especially valuable given that MOR eruptions have not been visually witnessed. I think that this ms has the potential to become a well-referenced classic. Their quantitative methods seem sound, albeit if I don't have the expertise to evaluate all the details (e.g., the tidal model).

We thank the reviewer for these positive comments.

I have comments and suggestions below. I do disagree with the authors' conclusion that intracrustal fluid release is needed to generate tephra- transporting megaplumes. Instead, I think their data suggests that lava flow heating is the energy source for the plume studied here.

Major comments:

1) More specific information on the observational constraints of megaplumes would be helpful (e.g., lines 31-35, lines 311-312). For example, How many megaplumes have been observed and with what instrumentation? Have volcanoclastic particles been observed in active megaplumes? What other evidence suggests that they correlate with volcanic eruptions? What is known about how long megaplume events last?

We fully agree these are helpful to highlight, and our revisions are as follows:

Added here details on the detection, instrumentation and characteristics of megaplumes and their association with eruptions:

"The detection of megaplumes along MORs by physico-chemical measurements in the water column has occurred both fortuitously during pre-planned surveys [2, 3, 8] and during rapid response cruises undertaken following the detection of geophysical evidence for submarine eruptions [9, 10], such as seismic or hydrophone activity (see [5] for a review). In the case of the former, subsequent ocean floor surveys, when conducted, have provided evidence

for contemporaneous eruptive activity [4, 11] and megaplume creation appears to be linked in space and time with deep sea volcanic events [5]. Observed concentrations of labile chemical species in megaplume fluids, such as H₂ [12] and dissolved Fe [13], generally indicate that the period of hydrothermal discharge must be relatively brief and that megaplume formation is therefore an ephemeral processes, probably associated with transient magmatic events.”

Revised caption to Fig 1 to state the number of megaplume events in parentheses after they are mentioned.

Additional note: Tephra has not been detected inside sampled fluid in the megaplumes that have been sampled to date - plausibly this is because the megaplumes have only been witnessed many weeks after their formation [5] and hence any tephra will have deposited beforehand.

2) The assessment of the source of the thermal energy for the megaplume could be expanded upon. The estimates given on lines 261-280 show that lava flows provide approximately the same amount of energy as is estimated for the megaplume, leading me to think lava flows are a very plausible energy source. I see that you say that only 30% of the lava thermal energy goes into the plume, however I am unconvinced by this brief statement. The lava will eventually cool to ambient temp and release its thermal energy. Because there is no mention of time in these calculations, I don't see why having a crust slow the rate of heat transfer is relevant to the calculation? One would need to estimate Q from the flow rather than the total amount of energy released. Even if 30% efficiency is right, the lava flow and plume energy estimates are still within a factor of 3, which seems fairly close to me given that there are other uncertainties that aren't fully explored here.

We have re-written this section and now, as suggested, take the approach of evaluating whether lava can provide the necessary heat flux rather than a simple consideration of the total energy contents. We are grateful to the reviewer for prompting us to assess this as we had not fully exploited this aspect of our results.

3) I would also appreciate more information about the nature of these intracrustal fluids. What are they? What do we know about them? Why are they suddenly released during a tephra producing submarine eruption? To me it seems arbitrary to invoke the release of additional fluids of unknown origin without a mechanism.

The pore fluid here referred to is interstitial water in the porous crust (as modelled by Lowel and Germanovich, 1995). In re-writing this section we reevaluated the possibility of *in situ* heating of pore fluids by the dyke (forming one of the primary theories of megaplume formation reviewed in our introduction; Lowel and Germanovich 1995) and - in light of the prolonged period of heating required - no longer present it as a feasible source of energy.

4) I think that the ms could be clearer about the observations and model assumptions regarding particle size. That is, did the Clague et al., study observe trends in particle size? Did you use a mean particle size from their observations? From the main text, I think that you only model one

particle size (that is, fall speed), however it is unclear. There is a section on polydisperse deposition in the supplement, however it was unclear how this connected to the assumptions and treatment in your model.

Clague et al. partitioned using sieving into 4 groups (now better noted, as described in first revision below). They did not discuss any trends in the data, focusing on qualitative description of the tephra and an estimate of the total tephra mass (now mentioned). We used the 250-500 micron range and a representative settling speed for this range of 3 ± 1 cm/s. To make these points clearer, we have made numerous revisions:

Added: in a new paragraph discussing model assumptions following Eqn. (2):

“In our analysis, we will choose a specific particle range (extracted by sieving [16]) and assume a particle settling speed representing this group.”

Added to text in first paragraph of section “Dispersal dynamics”:

“The pushcores were separated into size categories using sieving and the relative mass of tephra measured.”

Added below Eqn (a), a parenthetical clarification: “(or representative settling speed for a particle range under consideration)”

Added below Eqn (1): “Eqn. (1) provides the deposition field under the assumption of a single (or representative) particle settling speed w_s . However, deposits naturally contain a range of settling speeds. As discussed in section 3 of the Supplementary Methods, a general deposition can, in principle, be described in terms of a superposition of Gaussians. In our analysis, we will choose a specific particle range (extracted by sieving [16]) and assume a representative particle settling speed for this range.”

Revised in the context of determining a settling speed: “A representative settling speed for this group is determined as $w_s = 3 \pm 1$ cm/s.”

Added new paragraph in section 1 of the Supplementary Methods:

“In the analysis above, it has been assumed that the particle species is represented by the same settling speed w_s , while volcanic tephra will naturally involve a polydisperse distribution. As discussed in section 3, the deposition field resulting from a polydisperse distribution can, under our assumption of a dilute suspension (see below), be represented as an integral superposition of Gaussians of the form (5), in which the integrand is weighted by a mass distribution function representing the concentration of particles as a function of particle size d . To apply our inversion, we choose a group of particles in the size range $[d - \delta d/2; d + \delta d/2]$ and assign a representative settling speed for this range. The data of [10] was partitioned using sieving into four categories, and we use the particle range containing the most number of particles, 250-500 m, for our analysis. A representative range of settling speeds for this range is given by $w_s = 3 \pm 1$ cm/s (see section 4), which we use in our analysis.”

Added to section 3 of the Supplementary Methods to clarify purpose:

“In our analysis, we choose a representative settling speed w_s for the group of particles we consider ($d = 250\text{-}500\ \mu\text{m}$) and apply the monodisperse theory of Eqn. (6). It is instructive to consider how a polydisperse deposition can be predicted mathematically in order to uncover the strength of this approximation in a given situation.”

5) The characteristic length scale, L , of the deposit is central to your model. I didn't understand how L physically related to the actual Clague et al. deposits until reading the supplement (that is, fitting equation 2 to the tephra profile). I suggest you are more clear about the physical meaning of L in the main text. Some readers might think L is where you suddenly stop finding tephra (which is not the case).

Added below Eqn. (1): “In accordance with (1), the distance scale L encompasses ~93% of the mass of deposited tephra”

Added to caption of Fig. 4:

“defined as the decay scale of the Gaussian model of Eqn. (1) (and equal to the distance from the centre encompassing ~93% of the mass of tephra dispersed by the umbrella of the particle group under consideration)”

(quick note: it might be tempting to redefine L such that it encompasses a more rounded number like 90% but this introduces awkward prefactors in the clean theoretical definition $L = \sqrt{Q/w_s}$ and in the expression of Eqn. (1) which are better to retain)

6) A short statement related to your assumptions or treatment of the steadiness of the eruption could be helpful. Your model assumes that eruptions are steady, but in reality eruptions may not be. How does unsteadiness affect the ability to interpret deposits? How would one recognize a steady vs. unsteady eruption from the Clague et al. deposits?

Added: In text immediately after Eqn. (1):

“Another assumption underlying the model is that the plume is sustained by an approximately steady buoyancy source over the eruptive duration. While a waning input rate can be anticipated under the various theories for megaplume creation, we can anticipate that an approximation of a constant input rate provides a representation of the averaged properties of the plume system during the main period of energy release. Attenuation of the source energy will provide a relatively smaller contribution to the deposition at late times.”

Added to end of Results subsection:

“A reduction in the co-eruptive heat flux, as may occur due to a waning effusion rate, would result in a lower buoyancy flux at the base of the plume and therefore a decreasing rise height and deepening umbrella layer. Indeed, water column measurements in the region of a suspected megaplume above the Carlsberg Ridge in 2003 identified discrete peaks at different

depths, and of different magnitudes, in the physico-chemical parameters diagnostic of megaplume fluids [8], consistent with a varying heat flux at the source.”

Theoretically, time-dependence could be represented using superposition of the form:

$$\int_0^t m_0(t) \exp(-\pi v r^2 / Q(t)) dt ,$$

resulting in perturbation away from a pure gaussian (the extent of perturbation is dependent on the sharpness of variation of $m_0(t)$ and $Q(t)$). It is evident given the reasonable fit between our gaussian model and the data may indicate that a constant initial input was sufficient to characterise the NESCA deposition. More numerous data would be required to make more detailed inversion for a complete time dependence $Q(t)$, though we expect our inference to represent typical/mean values.

Line comments:

Lines 77 – 86: I think that authors are correct in using a model of a spreading gravity current. They nicely explain that this is different than what is typically used for subaerial plumes. I think a brief explanation of why they choose a different type of model (compared to subaerial plumes) would be helpful for the readers on these lines. That is, in what ways does the submarine setting differ from the atmospheric setting to motivate a different kind of model? What basic assumptions go into their choice of model? (I now see that they have this farther down in the ms).

In the new revision, we now clarify that there is no reason why the model should not apply in the subaerial context. Indeed, Volentik et al. 2010 comment on the limitations of standard inversion methods for the context of primarily radially dispersing subaerial eruptions due to the lack of any physical treatment of the buoyancy-driven flow of the umbrella in the model, producing grossly unphysical inferences of atmospheric transport processes (highlighted to us by Ref 4), reflecting the fact that the dominant process for horizontal dispersal due to buoyancy-driven flow in the umbrella is entirely neglected. Encouraged by these comments and under the suggestion of Ref 4, we now adjust the text to explain that our method is likely also to apply to this situation, i.e. that distinctions between submarine and subaerial at a physical level in terms of relative significance of cross flow is unnecessary and reflected in our newly revised text:

Revised to: “Although advection by buoyancy-driven flow is often considered in idealized prototypical fluid-mechanical analysis of tephra dispersal by subaerial eruptions [42, 43], it is neglected in standard methods for inverting tephra deposits in subaerial contexts, owing to the need for a new kind of mathematical model needed to account for it. The most standard models and inversion toolkits designed for subaerial eruptions [44, 45] account for the advection of settling particles in atmospheric crosswinds and diffusive atmospheric mixing, but neglect buoyancy-driven dynamics of the plume umbrella. In situations where this approach is applied to near-axisymmetric subaerial eruptions, the method infers unphysical values for the

atmospheric diffusivity [46], reflecting the fact that the dispersal in such cases is dominated by buoyancy-driven flow in the umbrella.”

Lines 55-56: I disagree that the model shows “direct evidence” of a connection. Rather, the model results show that the energy required for pyroclast dispersal are the same/similar to the energy required for megaplumes. To me, direct evidence of a connection would be the observation of volcanoclastic particles within megaplumes.

Agreed (“direct” indeed has a usage that refers to real-time observation). **Revised** to “conclusive”

I am curious if this has been reported previously in the literature?

No, tephra has not been detected inside sampled fluid in the megaplumes that have been detected to date, likely due to the time scale of ~weeks between generation and observation.

Line 57-58: I find the term “highly unlikely” somewhat dissatisfying. Instead, can you make quantitative comparisons of the energy discharge. 1TW (need a heat loss model for lavas to see if one can lose heat quickly enough)

This is an excellent suggestion (cf. main comment 2). We have now added new discussion utilizing our estimate of flux to quantitative estimations of heat transfer associated with the proposed theories for megaplume production - our constraint on energy transfer provides new inroads for this (as opposed to relying on inferences based on total heat, and uncertainties on eruptive duration). Our revisions for this are substantial , and so we refer to the marked up paper for this new discussion (3rd paragraph of section “Sources of megaplume energy” onwards). The discussion includes both estimates of cooling rates from erupted submarine lavas, utilising a heat conduction model for submarine lavas (Dechamps et al. 2010), and an estimate of the potential for energy transfer to be provided by exsolved CO₂.

Line 76: Would it be possible to specify the composition of the “volume flux.”

Added “of fluid” to be specific.

Lines 95-96: Maybe a statement on how Clague et al. knew that the volcanoclastic particles came from a singular and same eruption?

Added: “The lava flow is the only example to have occurred after the region was covered by sediment transported by the Missoula floods, and it is therefore known that the deposit of tephra originated from a single eruption.” (more details in Clague et al. 2009 referenced just before this.)

Line 108-110: I’m not sure I understand.

The referenced sentence here is “We propose instead that the characteristics of the observed tephra deposition suggest a dominant transport mechanism by advection within the umbrella of the plume”. It is not clear to us what may need to be revised here so we have left it unchanged.

Line 115: Thinning of mass, but what about particle size sorting?

Our intention at this juncture of the manuscript is just to give a general qualitative overview of the essential reduction in the rate of particle fallout (all sizes) with distance.

Line 140: Can you specify what you mean by “which”?

Added: “whether fallout from the stem or the umbrella is the more dominant process in a given situation”.

Line 148-149: what about geostrophic flow?

Geostrophic flow is a fluid-mechanical regime in which Coriolis forces balance driving stresses to leading order or, in other words, a regime with negligible fluid inertia in the rotating frame. It is therefore a regime of flow, as opposed to an individual contribution or process of the kinds we are listing here, e.g. large-scale deep-ocean currents, mentioned here, are close to geostrophic flow.

Figure 3: Add seawater entrainment. Label the pink part of the plume as a mixture of particles and warm fluid. Include an arrow in the central part of the plume to indicate upward flow. Perhaps put an arrow to indicate a downward particle settling below the plume. I don't think I understand tidal sway. Put figure 3 before figure 2.

Added “seawater entrainment” and a new label to indicate the plume as a mixture of particles and warm fluid. We have also now left out the tidal sway to avoid confusion. We have also added arrows to indicate the settling.

Regarding the ordering, we feel it is important to describe the observations early (Fig 2 showing the dispersal map) to clarify the basis for our theory, before illustrating the theoretically proposed configuration schematically (Fig. 3).

Line 160: Perhaps say “elongated deposit” rather than “near-linear plume” Line 197: Maybe reference how particle size comes into the settling speed. Also you state your assumptions about how the particles are coupled to the flow.

Changed as suggested, thanks.

Line 198: Can you be specific about what is in this volume flux (mixture of particles and fluid?)

Added the word “fluid” here to make this clear.

Line 202: Could this equation be tested by comparing dispersal distances for different size particles within a single eruption? What if Q is not constant in time?

In principle, yes it could be, assuming one has enough samples. We refer to our response regarding time dependence above.

Line 208: How is the dispersal scale, L , compared to measurements on the ground?

Our detailed inference of L based on regression to all the data immediately follows this.

Line 223: Does this fitting mean that you need more than one observational profile to identify a source vent. Or, could you ID a vent with just one profile? Maybe you could comment on how more than one profile is needed to assess whether currents play a role in dispersal (what errors would be associating your model with a profile that was advected by currents).

This is an interesting point. In principle (i.e. in a truly ideal world), technically just four data points are required to constrain the 4 fitting parameters: the two coordinates of the source vent and the dispersal scale L and accumulation scale Ω_0 (and the samples need not lie in linear profiles, though it will aid the regression to have a range of distances). Practically speaking, however, multiple data points are needed due to inherent scatter, and it is likely that linear profiles in different directions similar to those provided by the pushcores of Clague et al. 2009 may assist this. For discerning currents versus a quiescent environment, we can speculate that a single observed gaussian profile might still be sufficient to infer an axisymmetric deposition, since the deposition profile due to an umbrella advected by a current involves exponential decay (at least in the direction of the current, e.g. Johnson et al. 2015) as opposed to a gaussian and would be skewed in one direction. The closeness of our inferred centre to the true centre of the eruption and the lack of any obvious skew makes the plausibility of the effects of a sustained cross flow unlikely in our case. If one had a generalised theory that includes cross flow and one were to fit for the cross flow speed, then it is likely that it would regress to zero cross-flow speed because of the essential axisymmetry (leaving aside anything more quantitative relating to the dispersal profile).

Line 248-249: It is not clear to me how you infer an eruptive timescale from an eruptive flux (that is, how is volume estimate for this calculation?).

The volume estimate comes from Clague et al 2009 and is based on the bathymetric data. This is now clearly stated.

Line 280: I think that this language is too strong. Your results “suggest” fluid release, but I don’t think prove it.

Changed as suggested.

Line 282: A better definition of this crustal fluid could be helpful. That is, how do you define a hydrothermal reservoir vs. heated pore fluid?

As mentioned above, we have rewritten this section and no longer present the accelerated release of pore fluid hypothesis as a possible explanation.

Line 287: Can you tell me what these microbes are and why they are associated with hydrothermal fluid?

This part has been re-written to make this clear. The purpose of referencing the microbes is that it supports (at least a partial) origin for megaplume fluids from the crust:

Added: “A crustal origin for at least a portion of megaplume fluids is consistent with the presence of (crustally-derived) thermophilic microbes observed in plume fluids at the Gorda

Ridge in 1996 [64] and the high rates of energy transfer required for tephra dispersal suggests that these fluids may represent the dominant component of megaplumes.”

Line 291: Here you mention the speed of heat transfer. However, nothing previously in your analysis suggested anything about speed.

We refer here to *rapidity* of the heat transfer, which is represented by the constrained quantity Φ (the rate of energy transfer, or heat flux).

Line 355-356: I disagree with this interpretation. You show that data fits a model.

We understand the reference sentence to be: “We have shown that km-scale tephra dispersal in the deep ocean can be explained by transport in the umbrella of a co-eruptively produced megaplume”. We agree that the data fits our mathematical model based on buoyancy-driven dispersal with an energy flux consistent with the production of a megaplume, and our summary seems consistent with our finding here (the phrasing does not assert the conclusion but shows that our explanation is consistent with observations).

Line 429: Typo, It remains “to”?

Corrected.

Line 433: Replace “bed source” with “base.”

Changed as suggested.

Equation 5: I did not verify all the details in these equations. However, I like your approach. One comment is that this treatment doesn’t include the energy required to loft particles upward. That is, it is a one phase treatment of a plume. A sentence on the assumptions going into this model could be helpful as well as why you can neglect the multiphase nature of the plume in the stem. I don’t have expertise to evaluate the accuracy of the tidal model.

This is an interesting point. Single-phase models provide a good approximation for dilute particle concentrations of $< 10^{-3}$. In light of the referee’s comment, we decided to estimate the order-of-magnitude mass concentration of particles through different stages of the flow. Significant new details are provided in a new subsection of section 2 indicating a dilute concentration, and references provided in the main text as follows.

Added after model is first introduced (below Eqn. (1)):

“A further assumption underlying the model is that the presence of particles does not significantly impact the fluid flow. As discussed in section 2 of the Supplementary Methods, this is likely to be an excellent approximation for this application owing to the considerably larger proportion of plume fluid compared to particle mass.”

Added estimation of predicted concentration using the theory of section 2 of the Supplementary Methods: See new subsection titled “Estimating tephra mass and concentration”.

Added before final section to clarify applicability of this approach:

“and the particle concentration is small enough such as not to significantly affect the fluid dynamics”

I had a hard time following the reaching the umbrella section. I think the coupling of the fluid and particles is important and am glad this section exists. It might help if you describe better the physical meaning of the dimensionless group in this section. How does it relate to other parameters like the Rouse number that are commonly used for fluid particle settling? Why do you use this dimensionless parameter?

In the revision, we took the opportunity to make this section of the Supplementary Methods as clear as possible, revising it throughout to include more discussion and details. A summary of these revisions is provided here:

Added and front-loaded new explanation of dimensionless number:

“As a metric to assess the satisfaction of this condition, we define the umbrella dispersal number:

$$\Lambda = L/r_0$$

representing the ratio of the umbrella dispersal length scale L to the maximum stem radius r_0 . If $\Lambda \ll 1$, then the umbrella-driven dispersal considerably exceeds the maximum distance that can be dispersed by the stem, which is consistent with the former being the dominant process. Conversely, if $\Lambda \gg 1$, dispersal cannot extend beyond the stem radius and will be limited to settle below the plume stem.”

Added a new intermediate step represented by Eqn. (17).

Revised second half (now sub-sectioned) deriving an explicit expression for the proportion of particles reaching the umbrella [the new Eqn. (22) and new Fig. 8, and preceding equations], yielding a more direct derivation than our previous argument based on the size of the exponent, but more elegantly determining the emergence of the dimensionless parameter Γ .

Thank you for pointing out the possible connection. The number Γ depends on buoyancy/stratification (N) and buoyancy flux (F_0) specific to buoyancy-driven plume dynamics, measuring competition between vertical and horizontal buoyancy-driven plume dynamics (umbrella versus stem dispersal). We have been unable to establish a connection to the Rouse number, as the Rouse number, which is dependent on a background shear flow and its shear rate, as well as turbulence parameterizations near a wall (von Karman constants) absent in our analysis.

Responses to Reviewer 4

We thank the referee for the detailed and thorough review of our manuscript, which has led to significant improvements. We were pleased to read the referee's positive comments recommending that the paper fits the scope of the journal, merits publication, makes an important and novel point, and that they wished all manuscripts they received were of this standard.

We agree with all the suggestions and have made corresponding revisions. The comments relating to regression particularly have led to elegant simplification of our method/grid-search that is now fully inclusive of all the data and now fully documented in the main paper. Please find our responses below, including comments from the annotated PDF, and please find the newly revised material in the submitted manuscript marked in blue.

My main issues concern the: (i) lack of any quantified error analysis on the final fit (and partially as a consequence stemming from this, the lack of errors in the estimated values for other parameters), and (ii) the seemingly arbitrary rejection of one of the four profiles used. As this is the first time the proposed model is used I feel that the fitting errors should be presented in detail, making the inclusion of bad fits especially relevant (at least in the error analysis).

To clarify first an adjustment to our regression procedure stimulated by point (ii), we will respond to this point first:

Point (ii) (previously excluded profile 4):

To address the comment regarding the former exclusion of one of the profiles, we have now revised our regression procedure in such a way that all the data is now included. Previously, for a given assumed eruption centre x_c , we fitted an individual gaussian to each profile first (of the form $m_{0p} \exp(-\pi d^2/L_p^2)$, where d is the distance between the pushcore and an assumed trial centre (on which a full grid search was made) and L_p was a *profile specific* fitted dispersal scale ($p = 1, 2, 3$)- on doing a grid search over all trial centres x_c , we determined the unique position for which the fits for profiles 1-3 yield the same value of $L_1 = L_2 = L_3$ (such a unique position existed, as the search amounted to triangularization); the scatter in profile 4 had made an *individual* fitting for L_4 difficult to constrain and was thus omitted. However, under our new approach we conduct a *simultaneous* fit for all 4 fitting parameters. With this method, all the data is regressed simultaneously and profile 4 does not present any issue of convergence when included alongside the rest of the data, and there is no reason to exclude it. The optimal fit for all the data runs through the centre of the scatter within profile 4 with approximately as many points above as below (16 above, 14 below, Fig. 2B). The new fitting yields an estimate of $L = 4.9$ km (similar to our previous value of $L = 4.7$ km) and an optimal centre that is slightly S and 800 m to the W (Fig. 2A); thus, all our conclusions remain unaffected, but the method is now simpler and fully inclusive of all the data. This is a highly worthwhile improvement, and we

thank the reviewer very much for raising this comment which stimulated us to adjust the procedure.

Added a new description of our procedure at the beginning of the section “Results”. We refer here to the new/revised material shown in blue therein.

We also remark (responding to a PDF comment below) that there is a unique global minimum in the RMSE with a clear centre, as shown in the figure here showing the R^2 value produced for every trial centre over our grid search, creating a convex function with a unique maximum. The black circle shows the actual centre inferred from recent high resolution bathymetry of the site obtained by D. Clague, pers comm. This is clarified in the revised text.

Our revisions include an estimate of the sample error by conducting a bootstrapping analysis (we believe the essence of a suggestion of the referee in the PDF line comments (see below)), revealing a standard deviation of 0.4 km in the fitted value of L.

Point (i) (error quantification):

We now report our estimated errors, propagate these through to our final inferred values, and illustrate them visually in Fig 4 (details below).

Revised values to include estimated uncertainties in inferred quantities:

“Using the inferred dispersal scale of $L = 4.9 \pm 0.4$ km and the representative settling speed of the group $w_s \approx 3 \pm 1$ cm s⁻¹, we predict using the inversion formula derived above, $Q_{umb} = w_s L^2$, that the volumetric rate of growth of the umbrella was $Q_{umb} \approx (7.6 \pm 3.6) \times 10^5$ m³

$s^{-1} \approx (2.8 \pm 1.3) \text{ km}^3 \text{ hour}^{-1}$. The implied rate of heat transfer at the hydrothermal source predicted by Eqn. (2) is $\Phi \approx (5.5 \pm 3.3) \times 10^{15} \text{ J hour}^{-1}$ or $1.5 \pm 0.9 \text{ TW}$.”

Revised Fig. 4 to show ranges of uncertainty in plume volume and energy (includes the uncertainty in eruptive duration, which does not come into the uncertainties for the fluxes above, but does come into our inference that a megaplume was created to form the dispersal now given as an uncertainty of 10-20 hours (as opposed to 12 hours given previously) based on a more detailed consideration of lava effusion rates.

Revised figure 4A to include shaded areas covering uncertainties:

Revised caption of Fig. 4 to include:

“The other curves (grey) represent the minimal and maximal inferences that would apply for settling speeds of $w_s = 2 \text{ cm s}^{-1}$ and duration $\tau = 10 \text{ hours}$, and $w_s = 4 \text{ cm s}^{-1}$ and duration $\tau = 20 \text{ hours}$, respectively, covering ranges of uncertainty in these parameters. Shaded bands represent the inferred range of values based on the range of dispersal lengthscale $L = 4.9 \pm 0.4 \text{ km}$ determined by fitting the predicted Gaussian shape to the observed data for the NESCA eruption (Fig. 2).”

Added error quantification in the discussion of fits in the Results:

“There is consistent overall agreement between the data and the gaussian trends in each case, with 51% of the data lying above the model curve, and an R^2 value of -0.6.”

Added sampling error on the inferred fitting of the dispersal length scale L (see also response to PDF line comments below): “By conducting a bootstrapping analysis in which our minimization search is conducted on 10,000 cases of randomly replacement-resampled data, we obtain the standard deviation in the inferred value of L to be 400 m, providing an estimate of the sampling error.”

On a more unrelated note I think that the manuscript can be reorganised to better utilise the nature publication structure style (i.e. methods section at the end of the paper). I would argue against including the mathematical foundations of the paper as supplemental material as in my opinion it is necessary information to evaluate the study. These and several smaller points can be found in the attached PDF document.

We agree. In accordance with the formatting checklist for Nature Communications, we have now renamed this to “Supplementary Methods” (instead of Information) - indeed, our intention was that this would be downloaded with the paper to appear as one PDF directly after the paper in agreement with the referee. We now also refer to the specific sections of the Supplementary Methods throughout the paper.

Overall I believe that the paper fits the scope of the journal, has an important and novel point, potential impact, and merits publication. I wish that all papers that are sent to me to review were of this standard. However, I still believe that some aspects should be critically reviewed before publication. I hope that the authors find the points raised constructive.

We thank the referee for the positive comments.

Responses to PDF Line Comments:

Line 10: I would say a mathematical model. It might be my personal bias but I feel that lately the default expectation from just the word model is a numerical model.

Agreed, **Revised** as suggested.

Figure 1: I would suggest switching panels A and B and adding a box around the relevant area shown in Panel A to indicate its position on the global map.

Agreed and **revised**: We have swapped the panels and added some geographic details to help locate the map in panel B (on trying various options, we found that there is not enough room to clearly include a box also to show the location of A, however, the new added labels are sufficient to clarify the location).

Figure 2: The nature of the error shown here is not detailed in the caption. As it is constant on all figures I'm guessing that it is associated with the observations? In which case I would suggest instead to try adding the error as an area plot behind the scatter, to show the extent around the gaussian fit more clearly.. (e.g. sketch)

We agree with the referee's recommendation to show as background rectangles the error (15% error in quantifying the % of glass in the sieved pushcores reported by Clague et al. 2009) and appreciate the plot to illustrate it. However, the bars resulting from this ended up being almost identical to our markers (see figure to the right which show bars (blue) almost indistinguishable in appearance from filling the interior of the markers [it is necessary to zoom in to see the difference]). We have thus revised the Fig. so that marker size represents measurement error and now explain in the caption that the marker size represents the measurement error as above.

Added to caption of Fig. 2: "The 15% error in estimating the mass of glass particles [10] corresponds to the size of the circular markers."

Added to text discussing sources of scatter (page 11):

"Since ours is a continuum model for the statistically averaged deposition field, deviations between our model and the data are expected. Scatter in the data is potentially represented by syn- or post-depositional processes such as aggregation during settling, the effect of topography (particles falling unevenly on surfaces), statistical noise in the turbulent and particle dynamics, sediment displacement and/or bioturbation, combined with errors in the measurement of the proportion of tephra in each pushcore [16] (indicated by the size of the markers in Fig. 2B)."

Figure 2 caption in regard to the lack of fitting on profile 4: Are there any quantitative criteria used for this?

Revised: Having revised our regression method to include this profile, no data is excluded and this sentence has been deleted.

Line 110: There are cases where similar axisymmetric deposition has been observed for subaerial tephra deposition that could be cited at some point in the paper, i.e. Volentik et al. 2010, doi:10.1016/j.jvolgeores.2010.03.011

Line 120: I generally see the point made here but I wouldn't necessarily say that it is strictly because of the dominance of the crosswind, rather than the lack of appropriate tools to account for it, e.g. see Costa 2010, doi:10.1002/grl.50942.

We agree and have now highlighted the likely need to account for this in the subaerial context as well. Thank you for these suggested references. We have now revised the text to emphasise the physical similarity between submarine and subaerial (particularly for axisymmetric dispersal), as opposed to appealing to any possible differences in the relative significance of cross flow between the two applications we had previously. In the oceanic context, an upper bound on oceanic turbulent diffusivity of $\kappa = 10^{-3} \text{ m}^2/\text{s}$ can account for just 10s of metres of spreading, and so the need for buoyancy-driven dynamics is clear. Thank you for highlighting this - indeed Volentik et al. note that application of Tephra2 leads to grossly unphysical values of inferred atmospheric diffusivity, required to account for horizontal dispersal, meaning that our methodology could be equally applicable to that context and we would like to consider this in future.

Revised text in contrasting our methodology (page 6), including references to the two suggested papers:

“Although advection by buoyancy-driven flow is often considered in idealized prototypical fluid-mechanical analysis of tephra dispersal by subaerial eruptions [42, 43], it is neglected in standard methods for inverting subaerial tephra data, owing to the need for a new kind of mathematical model needed to account for it. The most standard models and inversion toolkits designed for subaerial eruptions [44, 45] account for the horizontal transport of particles via advection by atmospheric crosswinds and diffusive atmospheric mixing, but neglect the advection by buoyancy within the plume umbrella. In situations where this approach is applied to near-axisymmetric subaerial eruptions, the method infers unphysical values for the atmospheric diffusivity [46], reflecting the fact that the dispersal in such cases can be dominated by buoyancy-driven flow in the umbrella [47].”

Line 121: This is a bit of a repetition of the previous sentence, I think that they can be merged and perhaps expanded to discuss modelling efforts in subaerial eruptions in a more comprehensive way.

As suggested, we now draw a more direct contrast between methods of inversion that consider ash transport purely by an advection-diffusion equation in atmospheric transport to buoyancy-driven flow in the umbrella, requiring a flow model of the kind we apply (see revision above).

Line 157: Although I appreciate the point and the citation here, I would like to ask if the authors considered using reanalysis data to add some kind of an average cross flow against depth plot to further strengthen their argument? Similar to the one shown later for N. There is a comprehensive reanalysis data archive here.

We have followed the suggestion and used a reanalysis archive. Our aim here is to report a typical order-of-magnitude for deep ocean currents for this region. This is indeed reflected in typical maximal time-averaged speed (u, v components shown below at two depths) for this specific region, which we now report.

Revised the citation to a reanalysis database and reported maximal month-averaged speeds in this region:

“(month-long speed averages are $< 0.006 \text{ m s}^{-1}$ at 1500 m depth in this region [40])”

Line 220: I would still add it to the analysis to have a quantification of the unfitness... Or at least present some objective criteria for the exclusion, as it's 25% of the available data.

The method is now inclusive of all the data (see main comment above).

Line 223: I would like to see some quantitative presentation of the fitting and eruption center estimation methodology at this point. From what I understand the authors chose the solutions that minimize RMSE along with the relative difference in the distances of the three solutions? Was this carried out via something akin to a grid search methodology or by including an extra factor to a cost function and minimising it?

Revision: We now include full details on our estimation methodology included within the paragraph containing the new Eqn. (3) (see main comment above).

I feel that, in either case, the methodology could be used to also quantify other highly likely positions around the one chosen? It would be interesting to see the spatial extent of other highly likely solutions and whether the actual position is included (I'm guessing it is).

Revision: We now make clear that there is a unique global maximum in our discussion following Eqn. (3) (see figure above).

Also, since the available data are significant in number I would suggest separating a random sample (~20% of the data) to have a realistic final error analysis, and presenting and discussing the fitting error in a bit more detail to show a typical error of the methodology. This is also the reason why I would argue that including Profile 4 would be better in the analysis, especially as this is the first example of using the methodology. If by including Profile 4 solutions for the center position fail to converge I think there is justification in removing the data, but I still think that it should be presented and discussed.

Revision: Profile 4 has been included as suggested and, given the new method (see above) does not cause any failure of convergence; we have now conducted a bootstrapping analysis (see above), included the R^2 value in the text, propagate intervals in parameters through to our inferred umbrella flux and buoyancy source heat flux, and revised our Fig. 4 to show as bars the uncertainty (see response to main comment).

Is this supplementary information or an Appendix? Since this is a nature journal, I would suggest modifying the Methods section on the paper and reframing the SI as a methods section, which in Nature journals is expected to come after the paper in any case.

Indeed this was intended (see 3rd main comment above).

Relabeled to “Supplementary Methods”, conforming to the correct naming convention specified in the Nature Communications preparation checklist.

I would rephrase to "As an illustrative example" as "exemplary" has other connotations.

Revised as suggested.

REVIEWERS' COMMENTS

Reviewer #1 (Remarks to the Author):

I required minor revisions in my previous review and I am now satisfied by the way the authors treated my comments and modified the manuscript accordingly.

I have no further comments and recommend this work for publication.

Kind regards,
Mathieu Colombier

Reviewer #2 (Remarks to the Author):

I am satisfied with the revised manuscript, as it has addressed the issues I presented in my first review. Only one minor change: In line 35, ref [6] should be added to [2,3, and 8], as those megaplumes were discovered during a "pre-planned" hydrothermal exploration survey, not a response cruise.

Edward T Baker

Reviewer #3 (Remarks to the Author):

Pegler and Ferguson, "Rapid heat discharge during deep-sea eruptions generates megaplumes and disperses tephra"

Pegler and Ferguson have done well to address the comments on their original ms. I still believe this is a novel and well-written manuscript that advances our understanding of submarine ash dispersal and megaplumes. I support publication at this stage or following minor revisions. I have a small number of comments and questions listed below.

Minor Comments

Lines 63-66: If there is such a conclusive link, I still wonder why ash particulates are not observed within megaplumes that have been sampled. Do you have any comments on reasons for the absence of particulates observed within megaplumes?

Equation 3: Can you include a reference for this expression or describe how it was derived? In particular, Woods and Bush (1999) Dimension and Dynamics of Megaplumes, may be an appropriate reference. Or, if your expression differs from their formulation, it would be interesting to understand the difference. I see this explanation now in the supplement. However, I am curious how your formulation differs from past formulations (e.g., by Woods and Bush)?

Lines 323-324: Can you be more explicit about how you are estimating lava heat flux (e.g., show the expression you are using?).

Lines 345-366: This discussion of magmatic volatiles is speculative. First, the CO₂ contents mentioned by reference [61] are the highest found in MORBs and therefore not representative of most MORBs. In general, there is much more H₂O in magmas than CO₂, so I am confused about why you are invoking

CO₂ as a potential primary driver? Why not examine the volatile flux from H₂O rather than CO₂?

Figure 6 is blurry.

Overall, a very exciting contribution.

Reviewer #4 (Remarks to the Author):

I'm very happy to see that the authors were very thorough in the revisions considering the comments from all 4 reviewers. I'm happy to suggest accepting the manuscript for publication after a few very minor points:

Figure 1: I would enlarge Panel A so that the whole figure fills the page and the white space is minimised

Figure 6: The image in the review manuscript appears to be of very low quality (lots of compression noise)

Paragraphs breaks: Paragraphs tend to be very long, especially ones that include added sentences after the revision. The general rule of thumb is one idea per paragraph (expressed as the first sentence). As a suggestion I've listed some lines where I could see some easy paragraph breaks below:

Lines: 33, 43, 60, 70, 127, 274, 286, 320, 332, 357, 381, 401, 439, 526, 672

Best of luck with the final revisions and congratulations on the paper!

Reviewer #1 (Remarks to the Author):

I required minor revisions in my previous review and I am now satisfied by the way the authors treated my comments and modified the manuscript accordingly.

I have no further comments and recommend this work for publication.

Kind regards,
Mathieu Colombier

Thank you for the review.

Reviewer #2 (Remarks to the Author):

I am satisfied with the revised manuscript, as it has addressed the issues I presented in my first review. Only one minor change: In line 35, ref [6] should be added to [2,3, and 8], as those megaplumes were discovered during a "pre-planned" hydrothermal exploration survey, not a response cruise.

We have included [6] here as suggested.

Edward T Baker

Thank you for the review.

Reviewer #3 (Remarks to the Author):

Pegler and Ferguson, "Rapid heat discharge during deep-sea eruptions generates megaplumes and disperses tephra"

Pegler and Ferguson have done well to address the comments on their original ms. I still believe this is a novel and well-written manuscript that advances our understanding of submarine ash dispersal and megaplumes. I support publication at this stage or following minor revisions. I have a small number of comments and questions listed below.

Minor Comments

Lines 63-66: If there is such a conclusive link, I still wonder why ash particulates are not observed within megaplumes that have been sampled. Do you have any comments on reasons for the absence of particulates observed within megaplumes?

We provided a likely reason for this in our previous response to the reviewer's comment 1:

“Tephra has not been detected inside sampled fluid in the megaplumes that have been sampled to date - plausibly this is because the megaplumes have only been witnessed many weeks after their formation [5] and hence any tephra will have deposited beforehand.”

Equation 3: Can you include a reference for this expression or describe how it was derived? In particular, Woods and Bush (1999) Dimension and Dynamics of Megaplumes, may be an appropriate reference. Or, if your expression differs from their formulation, it would be interesting to understand the difference. I see this explanation now in the supplement. However, I am curious how your formulation differs from past formulations (e.g., by Woods and Bush)?

We have added: “(see section 1 of the Supplementary Methods)” to the sentence containing Equation 3 to make its origin clear.

Our Equation (3) relates flux at the top of the plume stem with energy flux, which is not derived in previous papers (the switchover from line to point-source is also a new aspect of plume theory represented by Equation (3)). Woods and Bush 1999 focus on the long-term geometry of the neutrally buoyant geostrophic (Coriolis-dominated) eddy resulting from a thermal (a finite parcel of heat), which differs from what we consider, namely, a purely buoyancy-driven umbrella model (Sparks *et al.* 1991) - in fact, the solution for the flow and shape profile of the radial plume is unimportant in our model, as the tephra dispersal profile given by Equation (6) follows from the volumetric flux constraint $Q = 2\pi rhu$ alone. A geostrophic balance of the kind considered by Woods and Bush will likely establish on the longer-term time scales after the eruption has taken place (over many Coriolis cycles), similar to the regime of an oceanic “Meddy” (noted also by d’Asaro *et al.* 1994), for which the Coriolis parameter is a dominant component of the model.

Lines 323-324: Can you be more explicit about how you are estimating lava heat flux (e.g., show the expression you are using?).

The estimate is based on output from a mathematical model of heat conduction [62], as opposed to a single expression; the output of the model we quote comes from the solution of the model given typical conditions for submarine lava, as reported in the referenced paper [62]. We have inserted the words “mathematical” here to help convey that it derives from the solution to a differential equation, as opposed to an analytical expression:

“For flow thicknesses of ≥ 2 -3 m, theoretical calculations based on a **mathematical** model of heat conduction predict that the first few days of cooling are characterized by a waning heat flux at the lava-water interface within the range of 10^3 - 10^4 W m⁻² [61].”

Lines 345-366: This discussion of magmatic volatiles is speculative. First, the CO₂ contents mentioned by reference [61] are the highest found in MORBs and therefore not representative of most MORBs. In general, there is much more H₂O in magmas than CO₂, so I am confused about why you are invoking CO₂ as a potential primary driver? Why not examine the volatile flux from H₂O rather than CO₂?

Our aim here is to show that the most liberal case of closed-system degassing of a CO₂-rich MOR magma cannot provide the necessary heat flux. To this end, we therefore use the highest observed CO₂ in MORB.

Although the magmas will contain more dissolved H₂O than CO₂, the (pressure-dependent) solubility of H₂O in silicate melts is much higher than CO₂ and basaltic melts with 'normal' volatile contents will not exsolve appreciable volumes of H₂O at these water depths (>3 km) due to the high pressure. The hypothesis that explosive basaltic eruptions might be driven by accumulated CO₂ is well established in physical volcanology (i.e. since Jaupart and Vergnolle, 1989) and is the leading model for what drives explosive eruptions in the deep ocean (e.g. Helo et al. 2011; Clague et al 2009).

Figure 6 is blurry.

This appears to have occurred in the post-processing of the manuscript file after our submission (the figure we will submit is high resolution).

Overall, a very exciting contribution.

Thank you for the review.

Reviewer #4 (Remarks to the Author):

I'm very happy to see that the authors were very thorough in the revisions considering the comments from all 4 reviewers. I'm happy to suggest accepting the manuscript for publication after a few very minor points:

Figure 1: I would enlarge Panel A so that the whole figure fills the page and the white space is minimised

We have modified the layout of the panels to reduce empty space.

Figure 6: The image in the review manuscript appears to be of very low quality (lots of compression noise)

This appears to have occurred in the post-processing of the manuscript file after our submission (the figure we will submit is high resolution).

Paragraphs breaks: Paragraphs tend to be very long, especially ones that include added sentences after the revision. The general rule of thumb is one idea per paragraph (expressed as

the first sentence). As a suggestion I've listed some lines where I could see some easy paragraph breaks below:

Lines: 33, 43, 60, 70, 127, 274, 286, 320, 332, 357, 381, 401, 439, 526, 672

We agree and have been through the paper to start new paragraphs at several points, including those suggested above.

Best of luck with the final revisions and congratulations on the paper!

Thank you for the review.